# Dysfunctional mitochondria trap proteins in the intermembrane space

Tamara Flohr [1], Markus Räschle [2] & Johannes M Herrmann [1]✉

## Abstract

The accumulation of mitochondrial precursor proteins in the cytosol due to mitochondrial dysfunction compromises cellular proteostasis and is a hallmark of diseases. Why non-imported precursors are toxic and how eukaryotic cells prevent their accumulation in the cytosol is still poorly understood. Using a proximity labeling-based assay to globally monitor the intramitochondrial location of proteins, we show that, upon mitochondrial dysfunction, many mitochondrial matrix proteins are sequestered in the intermembrane space (IMS); something we refer to as "mitochondrial triage of precursor proteins" (MitoTraP). MitoTraP is not simply the result of a general translocation block at the level of the inner membrane, but specifically directs a subgroup of matrix proteins into the IMS, many of which are constituents of the mitochondrial ribosome. Using the mitoribosomal protein Mrp17 (bS6m) as a model, we found that IMS sequestration prevents its mistargeting to the nucleus, potentially averting interference with assembly of cytosolic ribosomes. Thus, MitoTraP represents a novel, so far unknown mechanism of the eukaryotic quality control system that protects the cellular proteome against the toxic effects of non-imported mitochondrial precursor proteins.

**Keywords** Intermembrane Space; Mitochondria; Nucleolus; Protein Targeting; Ribosome
**Subject Categories** Membranes & Trafficking; Post-translational Modifications & Proteolysis

## Introduction

The mitochondrial proteome is a genetic mosaic of nuclear and mitochondrially encoded proteins (Morgenstern et al, 2017; Rath et al, 2021). The mitochondrial genome is tiny and encodes only for a small set of proteins, most of which are hydrophobic core subunits of the respiratory chain and the $F_1F_o$ ATPase (Ott et al, 2016). Only eight or 13 proteins are encoded on the mitochondrial DNA of yeast and human cells, respectively. All mitochondrially encoded proteins originate from the bacterial ancestors of mitochondria, but most bacterial genes were either transferred to the nucleus or lost, presumably to better coordinate gene expression in the eukaryotic cell (Allen and Martin, 2016) and to reduce the risk of mutations which are frequent in mitochondrial genomes, particularly in older individuals (Gupta et al, 2023; Ross et al, 2013).

Most mitochondrial proteins are encoded by nuclear genes and synthesized in the cytosol, from where they are imported as precursor proteins into mitochondria (Chacinska et al, 2009; Herrmann and Bykov, 2023). Proteins of the matrix and the inner membrane contain N-terminal matrix-targeting signals (MTSs or presequences) which direct the precursors through the protein-conduction channels of the TOM and TIM23 translocases in the outer and inner membrane (Araiso et al, 2019; Callegari et al, 2020; Sim et al, 2023). Owing to their characteristic structural features, presequences can be reliably detected by targeting prediction programs (Emanuelsson et al, 2007). They bind to distinct binding sites on subunits of the TOM and TIM23 complexes from where they are passed on to the Hsp70 chaperone of the matrix (called Ssc1 in yeast). To improve chaperone binding, the matrix Hsp70 chaperone is dynamically recruited to the TIM23 complex by Tim44, an essential component of the import motor (Mokranjac, 2020). ATP-driven cycles of Hsp70 binding than ratchet precursor proteins into the matrix (Okamoto et al, 2002).

Presequences form amphipathic helices with one hydrophobic and one positively charged surface (von Heijne, 1986). Subsequent to the translocation into the matrix, presequences are removed from most mitochondrial proteins by the matrix processing peptidase (Vögtle et al, 2009). Yeast cells contain ~900 mitochondrial proteins, from which 600–700 use the TIM23 import pathway (Morgenstern et al, 2017). While typical MTSs are found in most of these proteins, many mitochondrial ribosomal proteins (MRPs) use targeting signals which do not adhere to the characteristics of MTSs (Bykov et al, 2022; Woellhaf et al, 2014). Why many MRPs have unconventional signals is not known. The overall structures of MRPs are similar to those of protein subunits of the cytosolic ribosome: they are rich in positive charges and often contain a large content of alpha-helical structures in order to bind to the RNA core of the ribosome. It was speculated that these structural features of MRPs which often mimic MTS properties allowed mitochondrial targeting, so that the development of additional presequences during evolution was not necessary (van der Sluis et al, 2015).

The reliable targeting to and import into mitochondria of the 900 (yeast) to 1400 (human) precursor proteins is crucial for mitochondrial biogenesis and cellular proteostasis in general.

[1]Cell Biology, University of Kaiserslautern, Kaiserslautern, Germany. [2]Molecular Genetics, University of Kaiserslautern, Kaiserslautern, Germany.
✉E-mail: Hannes.herrmann@biologie.uni-kl.de

Non-imported precursors can pose a severe thread to cellular function (Nowicka et al, 2021; Wang and Chen, 2015; Weidberg and Amon, 2018) and are a hallmark of many diseases including neurodegeneration and cancer (Coyne et al, 2023; Haakonsen et al, 2024; Jackson et al, 2018; Sutandy et al, 2023). In order to prevent toxic effects of non-imported precursors, cells upregulate the capacity of chaperones and the ubiquitin–proteasome system in the cytosol upon mitochondrial dysfunction (Boos et al, 2019; Sutandy et al, 2023; Wrobel et al, 2015). In addition, they induce the facilitated sequestration of mitochondrial precursors therefore forming transient aggregates called MitoStores (Krämer et al, 2023; Zhou et al, 2011). Despite these defense mechanisms, non-imported precursor proteins can considerably compromise cellular fitness. However, the specific physiological consequences of non-imported precursor proteins are poorly characterized and it is not well understood where their toxicity arises from.

For this study, we used an unbiased proximity labeling approach to map the submitochondrial proteomes of the matrix and the intermembrane space (IMS). We thereby compared mitochondria of healthy cells to those of ATPase mutants which are defective in intramitochondrial ATP synthesis. Unexpectedly, we observed the accumulation of many matrix proteins in the IMS of the compromised mitochondria. In particular, MRPs were trapped in the IMS, many of which were proteins that lack typical MTSs. We next developed an in vitro assay based on the subcompartment-specific expression of viral proteases that allowed us to monitor the import of the model substrate Mrp17 into the IMS and matrix of mitochondria. Using this assay, we could show that matrix proteins with typical MTSs, such as Atp1, are not imported into dysfunctional mitochondria, however Mrp17 was efficiently translocated across the outer membrane and trapped in the IMS. If the transport of Mrp17 into mitochondria was prevented by fusion to an N-terminal GFP domain, Mrp17 was mistargeted to the nucleus where it associated with factors involved in the biogenesis of cytosolic ribosomes. Thus, we describe here a novel import pathway into mitochondria that, under conditions at which mitochondria are poorly energized, leads to the trapping of specific precursor proteins in the IMS potentially to prevent the toxicity of non-imported precursors outside of mitochondria.

## Results

### APEX-based sublocalization of the mitoproteome

The matrix and the IMS of mitochondria contain distinct sets of proteins to carry out their compartment-specific functions. Whether mitochondrial dysfunction affects the intramitochondrial distribution of proteins is not known. The budding yeast *Saccharomyces cerevisiae* is the best-characterized model to study protein translocation into mitochondria. The proteomes of the mitochondrial matrix and the IMS have previously been analyzed on the basis of biochemical fractionation experiments, in particular upon selective rupturing the outer membrane by hyperosmotic conditions or by employing proapoptotic factors (Morgenstern et al, 2017; Vögtle et al, 2012). An alternative strategy employed an engineered and improved version of ascorbate peroxidase (APEX2) as an excellent tool to biotinylate proteins with spatial and temporal control in subcellular compartments (Rhee et al, 2013).

Since their cell wall renders the budding yeast *S. cerevisiae* impermeable to biotin-phenol, we performed the labeling with mitochondria (Fig. 1A) that had been isolated from strains in which APEX2 was targeted selectively to the IMS or the matrix by fusion to subcompartment-specific targeting sequences (Appendix Fig. S1A,B). The accurate localization into the IMS and the matrix was confirmed by split-GFP reporters (Fig. 1B) and protease protection assays (Fig. 1C). Expression of these reporters did not compromise the mitochondrial protein import competence (Appendix Fig. S1C) or cellular growth rates (Appendix Fig. S1D). Addition of $H_2O_2$ resulted in a rapid (within 1 min) accumulation of streptavidin-bound proteins, revealing subcompartment-specific patterns of biotinylated proteins (Fig. 1D; Appendix Fig. S2A,B). Affinity purification on streptavidin beads followed by Western blotting confirmed the subcompartment-specific nature of the IMS-APEX- and matrix-APEX-based proteomic mapping approach (Fig. 1E,F).

Next, we used proteomics with wild-type cells expressing IMS-APEX, matrix-APEX or no APEX enzyme (mock) in four replicates after treatment with biotin-phenol and $H_2O_2$ for 5 min, purified biotinylated proteins on magnetic streptavidin beads and analyzed the recovered proteins by mass spectrometry (Fig. 1G). Both APEX reporters were highly specific: matrix proteins were predominantly biotinylated by the matrix-APEX enzyme and IMS and outer membrane proteins by the IMS-APEX enzyme (Figs. 1H,I and EV1). Inner membrane proteins showed a mixed behavior, reflecting their individual topology in the inner membrane (Fig. EV1C). The proteins Cym1 and Atp12 that are listed as IMS proteins in the SGD database (Wong et al, 2023) behaved like matrix proteins (Fig. 1J), and for both proteins experimental studies had documented the misdiagnosed IMS annotation and had proved their steady-state occurrence in the mitochondria matrix (Alikhani et al, 2011; Lefebvre-Legendre et al, 2001). Thus, the use of the APEX enzymes allowed the reliable and comprehensive mapping of the submitochondrial localization of proteins on a proteome-wide scale.

### Intramitochondrial protein distribution depends on mitochondrial function

Mitochondrial dysfunction impairs protein import efficiency and results in proteotoxic stress (Haakonsen et al, 2024; Merkwirth et al, 2016; Wrobel et al, 2015). However, whether mitochondrial dysfunction also affects the intramitochondrial localization of proteins is unknown (Fig. 2A). Mitochondrial defects can arise from mutations of the mitochondrial genome, a widespread phenomenon in aging human cells (Kauppila et al, 2017). We therefore used a mutant of the mitochondrial $F_oF_1$ ATPase in which the *ATP6* gene of the mitochondrial genome was replaced by *ARG8* that had been modified to adhere to the mitochondrial genetic code (Rak and Tzagoloff, 2009) (Figs. 2B and EV2A). This strain stably maintained its mitochondrial genome by selection on arginine-deficient media and synthesizes the mitochondrially encoded subunits of complex III and IV (Fig. EV2B). Even though it is unable to grow on non-fermentable carbon sources (Fig. 2C) it is less severely compromised than mutants lacking the mitochondrial DNA (Fig. EV2C) and maintains a moderately reduced membrane potential (Fig. EV2D,E). Since the complexes of the respiratory chain are expressed in this mutant and built up the inner

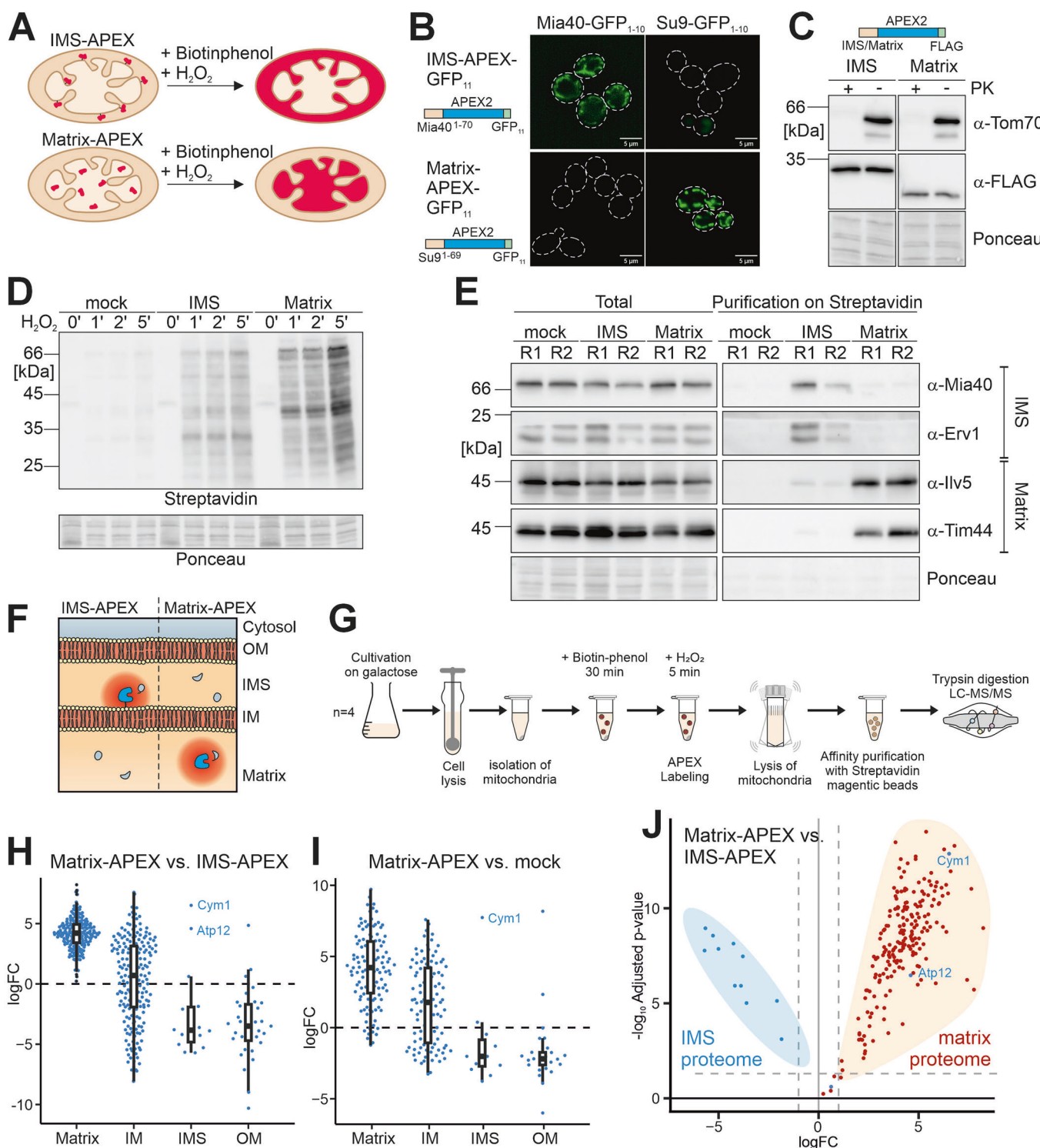

membrane potential, mitochondria of this mutant are fully competent to import proteins, albeit at reduced rates (Fig. 2D). We expressed IMS-APEX and matrix-APEX in this mutant and the sensitive split-GFP assay confirmed the reliable targeting to the IMS and matrix, respectively (Fig. EV2F,G; Appendix Fig. S3A).

We grew wild-type and Δatp6 cells to mid-exponential phase on the fermentable carbon source galactose, isolated mitochondria and

mapped the mitochondrial proteome by the APEX-based biotinylation assay (Fig. 1G; Appendix Fig. S4A). Whereas the proteome patterns found as background in the mock samples of wild-type and Δatp6 mitochondria (Appendix Figs. S3B and S4B) as well as the matrix proteomes of both cell types were largely identical (Appendix Fig. S4C), the protein compositions in the IMS strongly differed between healthy and compromised mitochondria: proteins which, in healthy cells,

◀ **Figure 1. Proximity-based mapping of the submitochondrial proteome.**

(A) Compartment-specific proximity labeling. (B) Yeast cells expressing a split-GFP reporter in the matrix (Su9-GFP$_{1-10}$) and the IMS (Mia40-GFP$_{1-10}$) were transformed with the indicated APEX fusion proteins. Fluorescent mitochondrial signal indicates colocalization. (C) Mitochondria isolated from the indicated strains were treated with proteinase K (PK) to show that the APEX enzymes are protease-inaccessible and thus located within the mitochondria; the outer membrane protein Tom70 served as a surface-accessible control. (D) Mitochondria of the indicated strains were H$_2$O$_2$-treated for the times indicated before biotinylated proteins were visualized with luminol staining using peroxidase-coupled streptavidin. (E) Mitochondria of the indicated strains were H$_2$O$_2$-treated for 2 min and lysed. Biotinylated proteins were isolated and subjected to Western blotting. (F, G) Proteomes of the IMS and the matrix were mapped by proteomics. Biotinylation patterns from $n = 4$ experiments allowed to distinguish the residents of the two mitochondrial subcompartments. (H–J) Relative enrichment (log$_2$ fold change, logFC) for the samples indicated were plotted for proteins residing in the matrix, the inner membrane (IM), the intermembrane space (IMS) and the outer membrane (OM), respectively. Boxes of the violin plots represent the data range from the first (Q1) to the third quartile (Q3), with the line in the middle representing the median. The minimum/maximum whisker values were calculated as Q1/Q3 ± 1.5 * interquartile range (IQR). Proteins that had been classified as matrix (red) or IMS (blue) are indicated; the matrix proteins Cym1 and Atp12 were previously reported as IMS proteins, which is incorrect (Wong et al, 2023). Significantly enriched proteins are indicated by the dashed lines (logFC > 1, P value < 0.05) (J). Source data are available online for this figure.

reside in the matrix accumulated in the IMS of the Δatp6 cells (Fig. 2E,F). Thus, the functional state of the mitochondria apparently left a characteristic footprint on the submitochondrial proteome. The IMS mislocalization of matrix proteins is not restricted to cells with mtDNA mutations but was also found in mutants of nuclear-encoded proteins which impair the ATPase activity such as in cells lacking the ATPase biogenesis factor Atp23 (Zeng et al, 2007) (Fig. EV3A–I). Apparently, reduced mitochondrial ATP levels induce the mislocalization of matrix proteins to the IMS. This IMS accumulation was not simply the result of a general import block of matrix proteins as the mislocalization affected only a specific subgroup of proteins, many of which were proteins of the small and large subunits of the mitochondrial ribosome (Fig. 2G,H; Appendix Fig. S4D; Datasets EV2 and EV3). Many mitoribosomal proteins (MRPs) have untypical targeting sequences and only score low in predictions programs for mitochondrial targeting such as TargetP (Bykov et al, 2022; Woellhaf et al, 2014). Low TargetP scores (Emanuelsson et al, 2007) were also frequently found among the IMS-localized MRPs (Fig. EV3F; Dataset EV2) (Bykov et al, 2022). Thus, mitochondrial dysfunction deviates a dedicated group of matrix proteins, in particular MRPs but also other matrix proteins with low TargetP score, into the IMS where they are presumably non-functional due to the absence of rRNA in the IMS (Fig. 2I).

## Mrp17 import occurs independent of ATP

Most matrix proteins carry N-terminal presequences, however, many MRPs lack such presequences but use internal targeting sequences instead (Bykov et al, 2022; Woellhaf et al, 2014). Why MRPs use such unconventional targeting signals is unknown. We used Mrp17 (bS6m) as a model protein of such a presequence-less protein to elucidate its import into isolated yeast mitochondria (Fig. EV4A). For better detection by autoradiography, we fused dihydrofolate reductase (DHFR) or a permanently unfolded DHFR mutant (C7S, S42C, N49C; DHFR$^{mut}$) (Vestweber and Schatz, 1988) to the C-terminus of Mrp17 (Fig. EV5A) (Bykov et al, 2022). When mitochondria were energized by preincubation with ATP and NADH, radiolabeled Mrp17 was efficiently imported into mitochondria, where it was protected against externally added proteinase K (Fig. 3A).

Efficient import of Mrp17 was dependent on Tom40, Tom20, Tom5 and Tim17, which are the core subunits of the protein translocases of the outer and inner membrane (Figs. 3A and EV4B,C) (Bykov et al, 2022). Moreover, the import of Mrp17

required the inner membrane potential and already low amounts of the ionophore carbonyl cyanide 3-chlorophenylhydrazone (CCCP) prevented Mrp17 translocation into mitochondria (Fig. 3B). To our surprise, however, depletion of mitochondrial ATP by incubation with the ATP-hydrolyzing enzyme apyrase did not block Mrp17 import (Fig. 3C). This was unexpected as the import of matrix proteins is generally assumed to occur in a strictly ATP-dependent reaction (Eilers et al, 1987; Voisine et al, 1999), and we also confirmed this ATP-dependence for other matrix-destined proteins such as Atp1 (Fig. 3C). The ATP-dependence arises from the essential role of the mitochondrial Hsp70 protein Ssc1 for matrix import which is the central, translocation-driving subunit of the import motor (PAM complex) (Kang et al, 1990). The motor-independent import of Mrp17 was confirmed by the observation that radiolabeled Mrp17 was efficiently imported into mitochondria that lacked functional Ssc1 (Fig. 3D). Likewise, Mrp17 import was independent of Hsp78 (Fig. EV4D) which can substitute for Ssc1 function to some extent (Schmitt et al, 1995). Protein import of Mrp17 only became ATP-dependent, once Mrp17 was fused to a tightly folded dihydrofolate reductase domain that was stabilized by the addition of methotrexate (Fig. EV4E). To further assess the motor-dependence of Mrp17 import, we generated a yeast strain in which Tim44 was depleted by CRISPR interference (Figs. 3E and EV4F,G). Tim44 is the essential central subunit of the mitochondrial import motor, crucial for Ssc1 association with the TIM23 complex (Schneider et al, 1994). Depletion of Tim44 did also not affect the import of Mrp17 into isolated mitochondria, whereas the import of Atp1 was clearly compromised (Figs. 3F,G and EV4H).

Next, we tested whether the import of Mrp17 into energy-deprived mitochondria can also be seen in vivo. To this end, we coexpressed Mrp17-mScarlet with Su9-neon green in cells of the cytochrome c oxidase-deficient Cox19 mutant (Fig. 3H). The presequence of the Neurospora crassa ATPase 9 subunit (residues 1-69) serves as well-characterized model MTS owing to its highly efficient mitochondrial targeting efficiency (Pfanner et al, 1987). Upon addition of oligomycin which inhibits the intramitochondrial ATP production, Su9-neon green was inefficiently imported into mitochondria, where the cytosolic signal increased with a higher concentration of oligomycin. In contrast, Mrp17-mScarlet was not detected in the cytosol (Fig. 3H). Thus, Mrp17 not only differs in the features of its matrix-targeting sequences, but also in respect to the energy dependence of the mitochondrial import reaction (Fig. 3G): ATP and the import motor are apparently not essential for their translocation across the outer membrane.

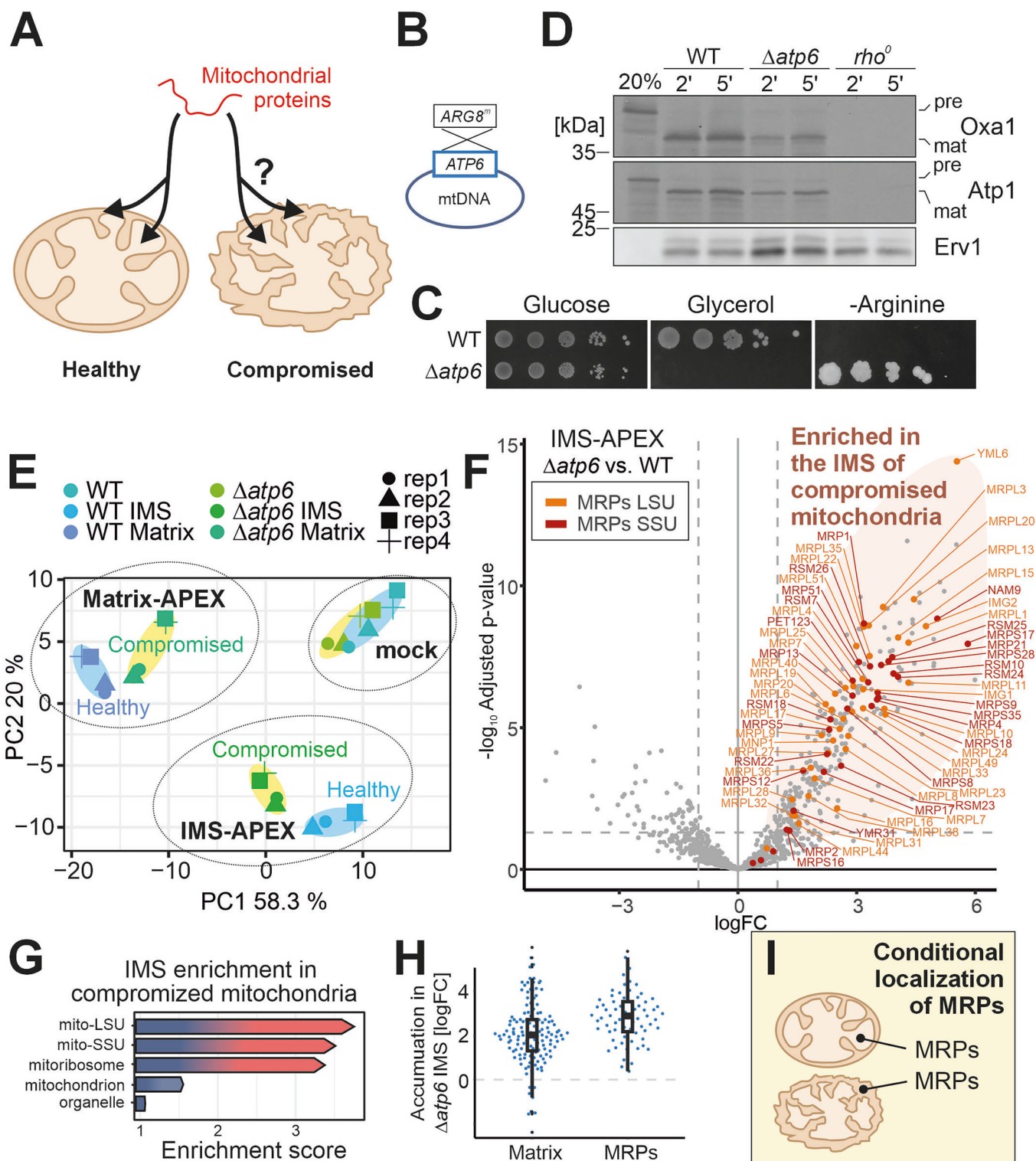

## MRPs are trapped in the IMS of dysfunctional mitochondria

Does the motor-independent import of Mrp17 direct the protein into the matrix or just into the IMS? To address this question, we imported radiolabeled Mrp17 into isolated mitochondria in the presence or absence of apyrase. Subsequently, we opened the outer membrane by hypotonic swelling before proteinase K was added (Fig. 4A). In the apyrase-treated samples, Mrp17 was degraded and hence present in the IMS, whereas Mrp17 was protease-resistant when the import occurred in the presence of ATP. This indeed suggested that Mrp17 was transported into the IMS when

**Figure 2.  Mitoribosomal proteins accumulate in the IMS of compromised mitochondria.**

(A, B) Mitochondrial function was compromised by the deletion of the *ATP6* gene on the mitochondrial genome. (C) Wild-type (WT) and *Δatp6* cells were grown to log phase in galactose-containing medium before tenfold serial dilutions were dropped onto medium containing glucose (with or without arginine) or glycerol as carbon sources. (D) Mitochondria of this strain maintain their competence to import radiolabeled precursor proteins of matrix proteins, albeit at reduced rates. Mitochondria from *rho[0]* cells which lack mitochondrial DNA show severe import problems of the matrix and inner membrane proteins Atp1 and Oxa1. Western blot signals of the mitochondrial protein Erv1 are shown as a control for equal loading of mitochondrial proteins. (E) Principal component analysis of the proximity-based mapping data of the proteomes of wild-type and *Δatp6* mutants showed that compromised mitochondria have an altered protein composition. (F–I) In the *Δatp6* mitochondria, many matrix proteins, in particular MRPs, were found to be biotinylated by the IMS reporter; the $\log_2$ fold change (logFC) values were calculated from $n = 4$ samples. Proteins of the small subunit of the mitoribosome are colored in red and of the large subunit in orange. Labeled are all mitoribosomal proteins that are significantly enriched (logFC > 1, *P* value < 0.05). (G) The proteins that were specifically enriched (logFC>1, *P* value < 0.05) in the IMS of *Δatp6* mitochondria were further analyzed by gene ontology (GO) enrichment, using the GOrilla tool (http://cbl-gorilla.cs.technion.ac.il/). (H) Relative enrichment scores (logFC) for the samples indicated were plotted for matrix proteins and MRPs, respectively. Boxes of the violin plots represent the data range from the first (Q1) to the third quartile (Q3), with the line in the middle representing the median. The minimum/maximum whisker values were calculated as Q1/Q3 ± 1.5 * interquartile range (IQR). Source data are available online for this figure.

mitochondrial ATP levels were low. In order to monitor the intramitochondrial location of imported proteins without mitochondrial subfractionation, we developed a reporter assay relying on two viral proteases that were expressed in the IMS and matrix of mitochondria, respectively (Fig. 4B). These proteases were derived from tobacco etch virus (uTEV3) and hepatitis C virus (HCVP) (Gao et al, 2018; Sanchez and Ting, 2020). None of the endogenous yeast proteins contains a cleavage site for these proteases and their expression in mitochondria did not impair cellular fitness (Appendix Fig. S5A). We generated radiolabeled fusion proteins consisting of Atp1 and Mrp17 and a segment with cleavage sites for these proteases, which we incubated with protease-containing mitochondria. Upon incubation of energized mitochondria containing the viral protease in the matrix resulted in complete processing of Atp1 and Mrp17 (Fig. 4C), indicating that both proteins were efficiently imported into the matrix. However, when ATP was depleted by apyrase treatment, Mrp17 reached a proteinase K-inaccessible intramitochondrial location, but was not processed by the matrix-targeted viral protease (Fig. 4D). Instead, it was cleaved when the viral protease was present in the mitochondrial IMS (Fig. 4E–G). The import of Mrp17 into the IMS still required the membrane potential across the inner membrane (Appendix Fig. S4B). Apparently, whereas the import of Atp1 was ATP-dependent, the translocation of Mrp17 across the outer membrane occurred independently of ATP. But if cells contained mitochondria of a low energetic state, Mrp17 became trapped in the IMS (Fig. 4F; Appendix Fig. S4C). This mistargeting to the IMS was also observed for other MRPs such as Mrpl40 and Mrpl28 (Appendix Fig. S5D) and was confirmed by protease accessibility assays with mitoplasts: once the outer membrane of *Δatp6* mitochondria was opened by hypotonic swelling, Mrp17, Mrpl36 and Mrpl40 became partially protease-accessible, confirming their localization in the IMS (Appendix Fig. S5E,F). Thus, the final intramitochondrial distribution of MRPs depends on the functional state of mitochondria (Fig. 4H).

## Trapping of MRPs prevents their accumulation in the nucleus

Why do cells trap non-imported MRPs in the IMS rather than accumulating them in the cytosol to allow their degradation by the ubiquitin–proteasome system? To assess the physiological consequences of the accumulation of MRPs in the cytosol, we expressed Mrp17 with an N-terminal green fluorescent protein

(GFP) thereby blocking its translocation through the TOM complex (Fig. 5A). Upon transient induction from a galactose-inducible promoter, this GFP-Mrp17 accumulated outside of mitochondria in one or two puncta per cell which often colocalized with nuclei (Fig. 5B,C). The N-terminal GFP apparently prevented mitochondrial import efficiently which was also evident from the observation that the GFP-Mrp17 fusion did not associate with mitochondrial rRNA, whereas the C-terminally tagged Mrp17-GFP fusion was bound to the mitochondrial 15S rRNA (Fig. 5D). We tested also GFP fusions to other MRPs and observed that also GFP-Mrpl40 and GFP-Mrpl28 showed a nuclear distribution; GFP-Mrpl28 thereby also was present in one or two foci per cell, whereas GFP-Mrpl40 was distributed throughout the nucleoplasm (Figs. 5E,F and EV5A,B). The nuclear location of GFP-Mrp17 was independent of its lysine residues as a version in which all lysines were replaced by alanines still formed nuclear foci (Figs. 5G and EV5C). However, whereas the overexpression of Mrp17 in mutants with mitochondrial dysfunction (*Δatp23*) was toxic and reduced growth rates even on non-fermentable carbon sources, the lysine-to-alanine version was well tolerated (Fig. 5H).

Intranuclear protein aggregates were observed before in other yeast studies, and the J domain protein Apj1 is known to specifically interact with such structures (den Brave et al, 2020; Kumar et al, 2024). We therefore analyzed the interactome of the nuclear chaperone Apj1 (Appendix Fig. S7A,B). When we prevented the mitochondrial protein import by expression of a competitive inhibitor (the $b_2$-DHFR clogger protein) (Boos et al, 2019), we found many MRPs among the Apj1-bound proteins as well as many proteins of the 90S ribosome assembly complex (Figs. 5I and EV5D; Appendix Fig. S7C; Dataset EV4). This confirms the potential of certain MRPs to become directed into the nucleus if their import into mitochondria is prevented and suggests (Fig. 5J), that their mistargeting to the nucleus induces—directly or indirectly—problems for the assembly of cytosolic ribosomes (Bassler and Hurt, 2019).

## Non-imported MRPs can associate with proteins of the 90S ribosome assembly complex

To identify interaction partners of the non-imported GFP-Mrp17, we expressed the protein for 5 h, purified it under native conditions on nano trap beads and subjected the isolated fractions to mass spectrometry (Figs. 6A; Appendix Fig. S6A,B). We found many proteins of the nucleolus in the GFP-Mrp17 interactome, in particular

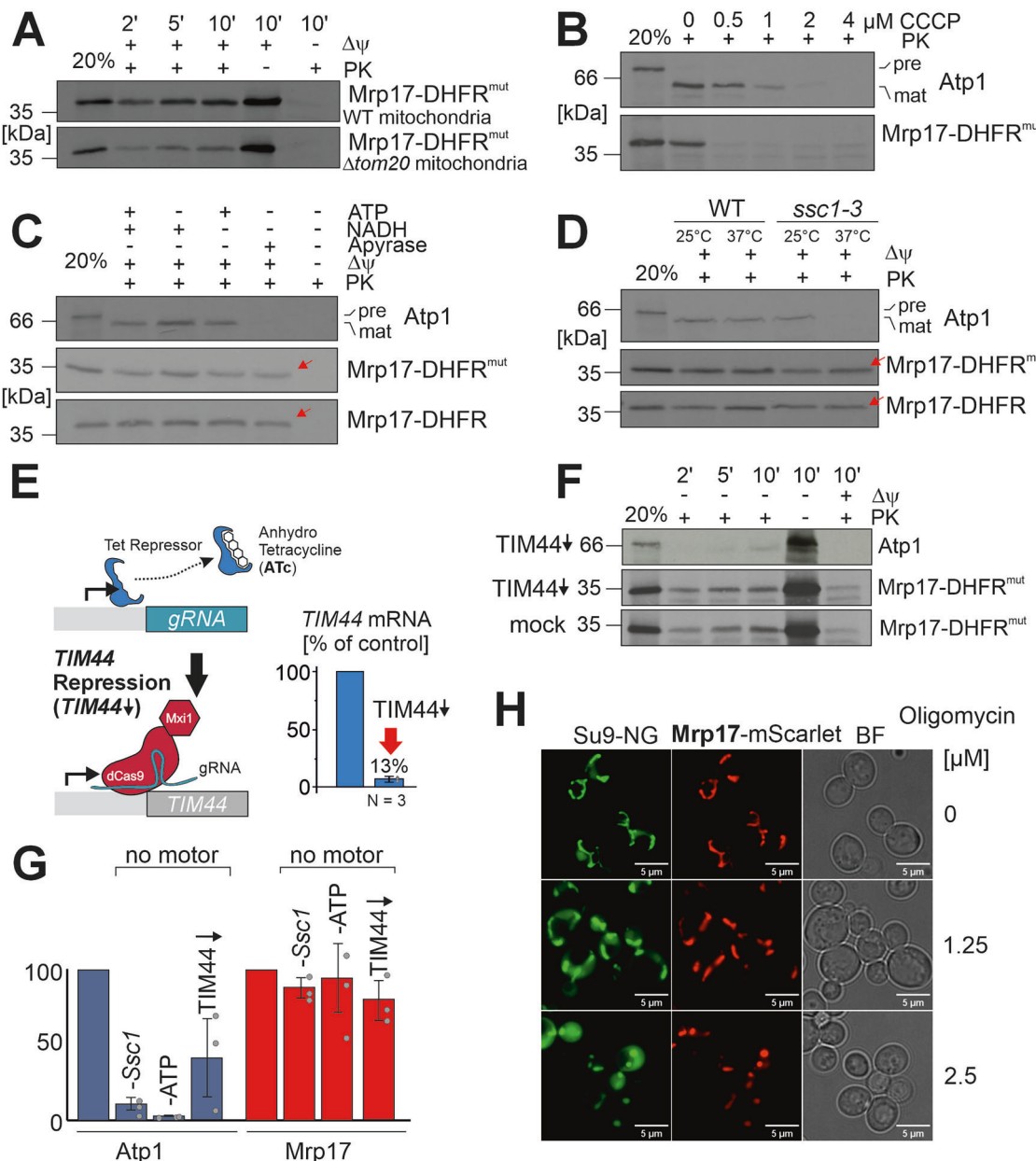

**Figure 3. Mrp17 is efficiently imported into ATP-depleted mitochondria.**

(A–D) Radiolabeled Mrp17-DHFR$^{mut}$, Mrp17-DHFR or Atp1 was incubated for the times indicated with mitochondria isolated from wild-type (WT) cells and the indicated mutants. In case of the temperature-sensitive *ssc1-3* cells and the corresponding wild type (D), mitochondria were pretreated for 10 min at 37 °C before the experiment. Non-imported protein was removed by treatment with proteinase K (PK). To dissipate the membrane potential (−Δψ), a control sample was treated with a mix of valinomycin, oligomycin and antimycin A. The 20% sample shows one-fifth of the radiolabeled protein that was used per time point of the import reaction. (E) The expression of TIM44 mRNA was strongly diminished upon CRISPR interference (CRISPRi) with a *TIM44*-specific guide RNA (Smith et al, 2016). *TIM44* transcript levels were quantified by quantitative PCR from three biological replicates. Shown are mean values and standard deviations. (F) Radiolabeled precursor proteins were incubated with mitochondria of the indicated strains and treated as described for (A). (G) The import efficiencies of Atp1 and Mrp17 into mitochondria of wild-type cells or the indicated mutants were quantified from three independent replicates. Shown are mean values and standard deviations. (H) The matrix-targeted Su9-neon green (Su9-NG) and Mrp17-mScarlet were constitutively expressed in Δ*cox19* cells. The indicated concentrations of oligomycin were added for 4.5 h. BF bright field. The fluorescence levels of the Su9- and Mrp17-fusion proteins were quantified. Source data are available online for this figure.

proteins of the 90S pre-ribosome complex (Fig. 6A,B; Dataset EV5). Almost the full complement of subunits of this assembly complex were co-isolated with GFP-Mrp17 (Fig. 6C). Thus, the non-imported GFP-Mrp17 is targeted into the nucleus where it interferes with the biogenesis machinery for 80S ribosomes.

When we purified the GFP-Mrp17 from cells, we found decent amounts of co-isolated 35S rRNA in the GFP-Mrp17 fractions, which represents the rRNA precursor of the 90S ribosome assembly complex. The 35S rRNA levels were lower than in pulldowns with the GFP-tagged 90S assembly factors Pwp2 and Utp18 but higher

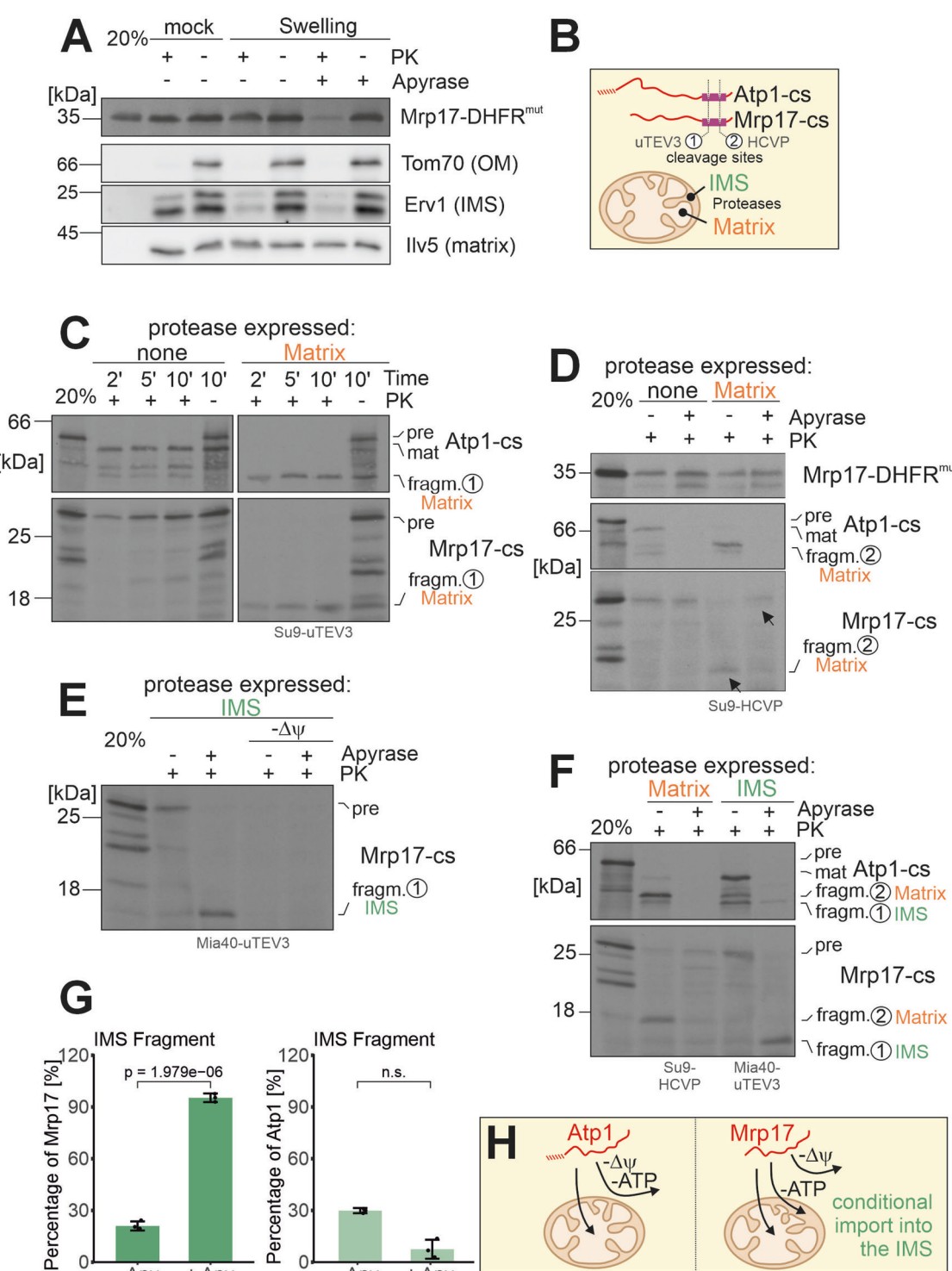

**Figure 4. Mrp17 is targeted into the IMS of de-energized mitochondria.**

(A) Mitochondria were incubated with ATP and NADH, or alternatively with apyrase, before radiolabeled Mrp17-DHFRmut was added for 10 min. Mitochondria were incubated in isosmotic (mock) or hypo-osmotic swelling buffer in the absence or presence of proteinase K (PK). The western blot signals of Tom70, Erv1 and Ilv5 were shown as markers for proteins of the outer membrane (OM), the IMS and the matrix. (B) Scheme of the viral proteases uTEV3 and HCVP, that were expressed in the IMS and matrix of mitochondria, respectively. (C–F) In vitro import experiments of Mrp17 and Atp1 fusions with a C-terminal segment harboring uTEV3 and HCVP cleavage sites and mitochondria containing the respective proteases, as indicated. Mitochondria were isolated from wild-type (WT) cells expressing Su9-uTEV3 (Matrix, (C)), Su9-HCVP (Matrix, (D, F)), and Mia40-uTEV3 (IMS, (E, F)). (G) The signal intensities of the protein fragments derived from cleavage by the IMS-located protease were quantified from three independent experiments. Statistical significance was calculated by a one-sided T test. Mean values and standard deviations are plotted. (H) Schematic representation of the conditional targeting of Mrp17 into the IMS under ATP-depleted conditions. Source data are available online for this figure.

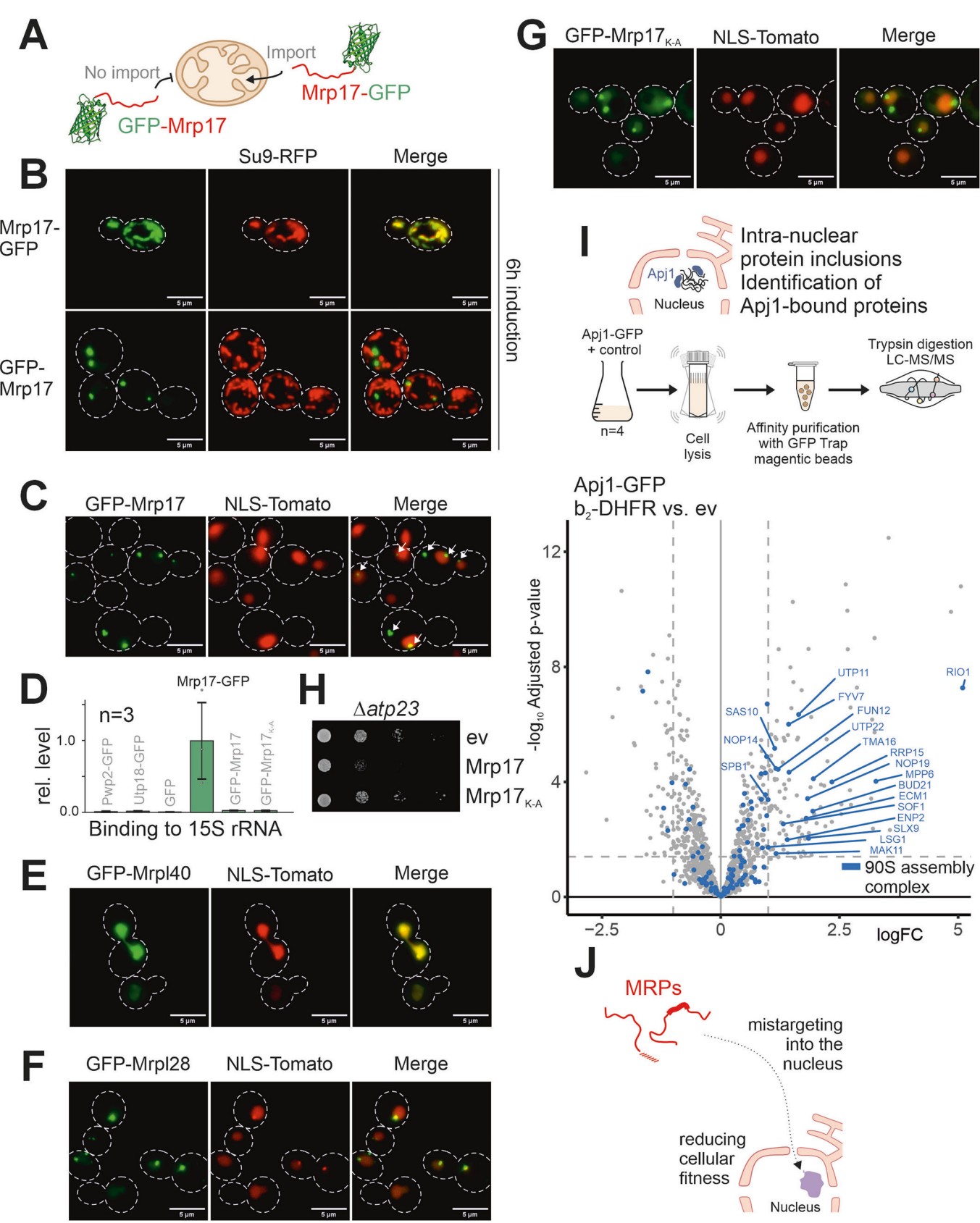

◄ **Figure 5. MitoTraP prevents the interference of Mrp17 with the assembly of the 90S pre-ribosome complex in the nucleus.**

(**A–C**) Mrp17-GFP and GFP-Mrp17 were expressed from an inducible *GAL10* promoter for 5 h and visualized by fluorescence microscopy. Mitochondria were stained by expression of Su9-RFP. The nucleus was stained by expression of NLS-Tomato. The N-terminal GFP prevents mitochondrial targeting of Mrp17 which accumulates in defined extramitochondrial foci that localize with the nucleus. (**D**) The indicated proteins were expressed in wild-type cells for 5 h at 30 °C. Cells were lysed in Triton X-100, and GFP fusion proteins were isolated on GFP-Trap-magnetic beads. Bound 15S rRNA was amplified from the samples and quantified in relation to the amount purified with Mrp17-GFP. Mean values and standard deviations of three independent experiments are plotted. (**E–G**) GFP-Mrpl40, GFP-Mrpl28 and GFP-Mrp17$_{K-A}$ were expressed and visualized as described for (**A**). (**H**) Mrp17 and the Mrp17$_{K-A}$ mutant were expressed from a constitutive *TPI* promoter in *Δatp23* cells. Cells containing an empty vector (ev) served as controls. Tenfold serial dilutions were dropped onto a glucose-containing plate and incubated for 2 days at 37 °C. (**I**) Apj1-GFP was constitutively expressed in wild-type cells. Mitochondrial protein import was competitively inhibited for 5 h by expression of the clogger (b$_2$-DHFR); a mock control contained an empty vector. Apj1-GFP was purified from whole cell extracts using Triton X-100 as a detergent on GFP-Trap-magnetic beads. Apj1-bound proteins were analyzed by mass spectrometry. The log$_2$ fold change (logFC) values were calculated from $n = 4$ samples. Proteins of the nucleolar 90S ribosome assembly complex are highlighted and significantly enriched proteins (logFC > 1, $P$ value < 0.05) are labeled. (**J**) Schematic representation of MRP precursors that are mistargeted to the nucleus. Source data are available online for this figure.

than in the non-toxic GFP-Mrp17 lysine-to-alanine mutant (Fig. 6D). The interaction of GFP-Mrp17 with the ribosome assembly complex was unexpected as Mrp17 has homologs in mitochondria and bacteria, but no direct counterpart in the cytosolic ribosome of eukaryotes.

To further test whether non-imported MRPs can interact with 90S complex components, we purified Pwp2-GFP- and Utp18-GFP-containing complexes from cells in which the mitochondrial import was competitively inhibited by the expression of the mitochondrial import clogger for 5 h and analyzed their content by mass spectrometry (Fig. 6E; Appendix Fig. S7A,B,D; Dataset EV4). These complexes contained the complement of established 90S complex subunits and, in addition, several MRPs. Thus, if the import of MRPs across the outer membrane is prevented, at least some of these proteins are targeted into the nucleus where they can interfere with the ribosome assembly. Whether this interference is caused by a direct incorporation of MRPs into the 90S complex or by a more indirect effect on nuclear proteostasis will have to be further characterized in the future. However, the overall structural similarity of many MRPs with subunits of the 80S ribosome (Greber and Ban, 2016) and the fact that cells transport proteins with helical, positively charged sequences of RNA-binding segments into the nucleus might explain the observed toxic potential of MRPs.

## Discussion

Eukaryotic cells face the challenge of assembling two completely distinct types of ribosomes, one in the nucleus/cytosol and one in mitochondria. Each ribosome species consists of about 80 protein subunits, and these roughly 160 different proteins are initially synthesized in the cytosol. Here we show that subunits of the mitochondrial ribosome can erroneously associate with components of the assembly complexes of the cytosolic ribosome, if their import into mitochondria is prevented. The observations reported here suggest that cells developed a strategy to direct MRPs into the mitochondrial IMS in energetically compromised mitochondria to avoid this misincorporation (Figs. 6F and EV5E). Thereby, the formation of non-productive aberrant assembly intermediates might be avoided. The extramitochondrial accumulation of MRPs has frequently been reported in cancer cells where they are indicative for unfavorable prognosis for unknown reasons (Kim et al, 2017; Pu et al, 2017; Zhou et al, 2024).

The feature of MRPs to be re-routed to the IMS might explain the occurrence of non-canonical targeting sequences that characterize this group of proteins (Bykov et al, 2022; Vögtle et al, 2009). Interestingly, not only may MRPs harm the assembly of cytosolic ribosomes, but ribosomal proteins—once targeted to mitochondria—can strongly interfere with mitoribosome assembly; a recent study showed that some ribosomal proteins contain import-avoidance signals at their N-termini to prevent their import into mitochondria (Oborska-Oplova et al, 2025). Thus, the assembly of two distinct ribosomes within eukaryotic cells is apparently a challenging problem as the two populations of ribosomal proteins can jeopardize cellular fitness if they are not accurately separated. During eukaryotic evolution, the processes of ribosome assembly have been segregated into two distinct compartments, the nucleus and mitochondria, potentially to reduce the risk of protein misincorporation (Martin and Koonin, 2006).

We named this IMS trapping of MRPs <u>Mito</u>chondrial <u>Tria</u>ge of <u>P</u>recursor proteins (MitoTraP) and assume that these IMS-located translocation intermediates will finally be degraded, potentially by the IMS-localized i-AAA protease Yme1 (YME1L in humans), which is a very powerful degradation machine of high physiological relevance (MacVicar et al, 2019). MRPs are not the only proteins that are trapped in the IMS by MitoTraP. Also, a number of other matrix proteins reach the IMS in dysfunctional mitochondria (Datasets EV1 and EV3). This is consistent with observations from in vitro import experiments in which some matrix proteins were found to reach the IMS in poorly energized mitochondria, but this phenomenon has not been systematically analyzed before (Longen et al, 2014; Peker et al, 2023; Rassow and Pfanner, 1991). It will be interesting to elucidate the structural features of precursors that determine whether they are trapped in the IMS of de-energized mitochondria or remain in the cytosol.

The presence of non-canonical targeting signals in MRPs is well known in the field. The MitoTraP model can now for the first time explain why MRPs have such signals. These non-conventional signals might avoid the association of the protein translocases of the outer and inner mitochondrial membrane. During translocation of matrix proteins with conventional matrix-targeting signals, the TOM and TIM23 complexes make direct contact, thereby forming continuous protein-conducting supercomplexes in contact zones of the outer and inner membranes (Chacinska et al, 2005; Zhou et al, 2023). Owing to the close cooperation of the translocases in both membranes, the matrix-located import motor (PAM complex) can drive the translocation of precursors across both membranes

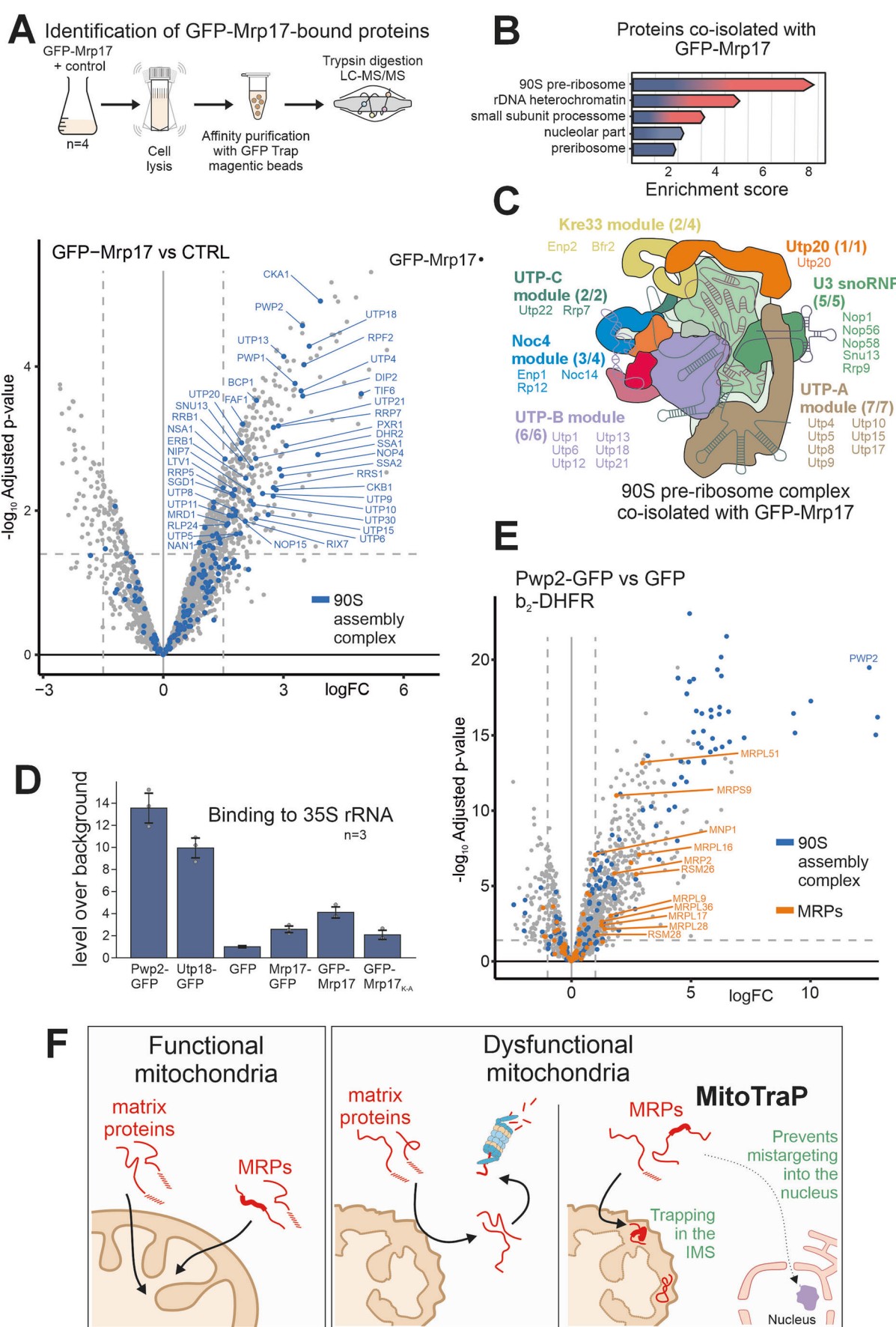

**Figure 6. MitoTraP represents a novel protein translocation pathway into mitochondria.**

(A) GFP-Mrp17 was expressed for 5 h in wild-type cells. GFP-Mrp17 was purified from cell extracts and co-isolated proteins were analyzed by mass spectrometry relative to proteins from a control sample (GFP). The $\log_2$ fold change (logFC) values were calculated from $n = 4$ samples. Proteins of the nucleolar 90S ribosome assembly complex are highlighted, and significantly enriched proteins (logFC > 1.5, P value < 0.05) are labeled. For details, see Dataset EV5. (B) Enrichment of functional groups that were co-isolated with GFP-Mrp17. The proteins that were specifically enriched (logFC > 1.5, P value < 0.05) in the GFP-Mrp17 pulldown were further analyzed by gene ontology (GO) enrichment, using the GOrilla tool (http://cbl-gorilla.cs.technion.ac.il/). (C) Schematic representation of the 90S pre-ribosomal complex of the nucleus (Hurt et al, 2023). Indicated are proteins of the different modules of the ribosome assembly complex that were found in the GFP-Mrp17 interactome. (D) The indicated proteins were expressed in wild-type cells, purified and co-isolated 35S rRNA was quantified by RT-qPCR. Shown are mean values and standard deviations from three biological replicates representing the 35S rRNA levels in comparison to those in the GFP control sample. (E) Proteins associated with a GFP-tagged version of the ribosome biogenesis factor Pwp2 were identified by mass spectrometry in extracts from clogger-inducing wild-type cells. Extracts from GFP-containing cells served as controls. The $\log_2$ fold change (logFC) values were calculated from $n = 4$ samples. MRPs and proteins of the ribosome assembly complex in the nucleus are highlighted. Significantly enriched MRPs (logFC > 1, P value < 0.05) are labeled. (F) Schematic representation of the intramitochondrial trapping of mitochondrial precursors in the IMS: Mitochondrial Triage of Precursor proteins (MitoTraP). Whereas some matrix proteins remain in the cytosol to be degraded, MitoTrap prevents the accumulation of mitochondrial precursor proteins in the cytosol, by trapping proteins in the IMS and thereby reducing the risk of protein interference with cellular activities. Source data are available online for this figure.

(Callegari et al, 2020; Chacinska et al, 2010). However, when import fails due to low ATP levels, proteins are not imported and accumulate in the cytosol. The non-conventional signals of MRPs might prevent the TOM-TIM23 association to allow trapping of the protein in the IMS, however, the mechanistic details of the translocation of MitoTraPed proteins into the IMS will have to be characterized in the future. It appears likely that their translocation mode differs from resident IMS proteins which either use bipartite presequences or embark on the MIA import pathway that depends on the oxidation of internal cysteine residues (Edwards et al, 2020; Riemer et al, 2009). The translocation of Mrp17 still required the membrane potential across the inner membrane, suggesting that the electrophoretic transfer of positively charged residues to the negatively charged matrix face of the inner membrane drives the translocation across the TOM complex upon ATP depletion. Since MRPs are characterized by many positively charged residues along their sequence, they might constitute the predominant clients of this novel targeting route into mitochondria.

# Methods

### Reagents and tools table

| Reagent/resource | Reference or source | Identifier or catalog number |
|---|---|---|
| **Chemicals, enzymes, and other reagents** | | |
| iTaq Universal SYBR Green Supermix | BioRad | 172524 |
| Apyrase from potatoes | Sigma-Aldrich | A6535 |
| Protease Inhibitor (cOmplete Tablets) | Roche | 04 693 159 001 |
| RNAse inhibitor Murine | NEB | M0314L |
| Pierce™ Streptavidin magnetic beads | Thermo Scientific | 815-968-0747 |
| GFP-Trap Magnetic beads | Chromotek | Gtma-20 |
| Sequencing grade trypsin | Promega | V5111 |
| Biotinyl tyramide | Cayman chemical | 27997 |
| Hydrogen Peroxide solution | Sigma-Aldrich | 216763 |
| Trolox | Thermo Scientific Chemicals | 10782831 |

| Reagent/resource | Reference or source | Identifier or catalog number |
|---|---|---|
| Sodium azide | Merck | 8.22335.0100 |
| L (+) Ascorbic acid | AppliChem | A3604 |
| MitoTracker Red CMXRos | Invitrogen by Thermo Fisher Scientific | M7512 |
| **Software** | | |
| MaxQuant software | | |
| R studio | | |
| ImageJ (Fiji) | | |
| Corel Technical Suite | | |
| Leica LasX | | |
| ImageLab by BioRad | | |
| **Other** | | |
| SPECTROstar$^{Nano}$ | BMG LabTech | |
| FastPrep-24 5 G homogenizer | MP Biomedicals, Heidelberg, Germany | |
| Spectrophotometer/ Fluorometer DS-11 FX+ | DeNovix | |
| CFX96 Touch Real-Time PCR Detection System | BioRad | |
| Dmi8 Thunder Imager | Leica | |
| CLARIOstar | BMG Labtech | |
| ChemiDoc MP, Imaging System | BioRad | |
| RNeasy Mini Kit | Qiagen | |
| RNase-Free DNase Set | Qiagen | |
| qScript cDNA Synthesis Kit | Quanta Biosciences | |
| TNT® Quick Coupled Transcription/Translation Kit | Promega | L2080 |
| Luna Universal One-Step RT-qPCR Kit | NEB | E3005E |

### Strains and growth conditions

The yeast strains and plasmids used in this study are described in detail in Appendix Tables S1 and S2, respectively. Biological

materials used in this study are available from the authors upon request. Unless specified, all strains were derived from W303 MATα (*leu2-3,112 trp1-1 can1-100 ura3-1 ade2-1 his3-11,15*).

The strains were grown at 30 °C either in yeast complete medium (YP) containing 1% (w/v) yeast extract, 2% (w/v) peptone and 2% (w/v) of the respective carbon source or in minimal synthetic respiratory medium containing 0.67% (w/v) yeast nitrogen base and 2% (w/v) of the respective carbon source.

## Growth assays

For spot analysis, the respective yeast strains were grown in liquid media. Yeast cells equivalent to 0.5 $OD_{600}$ were harvested at the exponential phase. The cells were washed in sterile water, and 3 µl of tenfold serial dilutions were spotted on the respective media followed by incubation at 30 °C.

Growth curves were performed in a 96-well plate, using the automated SPECTROstar$^{Nano}$ (BMG LabTech). The growth curves started at 0.1 $OD_{600}$, and the $OD_{600}$ was measured every 10 min for 72 h at 30 °C. The mean of technical triplicates was calculated and plotted in R.

## Cell lysates

For whole cell lysates, yeast strains were cultivated in liquid media to mid-log phase. 2 $OD_{600}$ were harvested by centrifugation (17,000 $\times g$, 1 min) and resuspended in 100 µl reducing loading buffer. Cells were transferred to screw-cap tubes containing 1-mm glass beads. Samples were boiled at 96 °C for 5 min. Cell lysis was performed using a FastPrep-24 5 G homogenizer (MP Biomedicals, Heidelberg, Germany) with three cycles of 20 s, speed 6.0 m/s, 120 s breaks, glass beads. Lysates were stored at −20 °C until further use. Equal amounts were resolved via SDS-PAGE.

## Antibodies

The antibodies against Mia40, Erv1, Ilv5, Tom70, Mam33, Mrpl40, Mrp17, Mrpl36, Mrp10 and Tim44 were raised in rabbits using recombinant purified proteins. The FLAG antibody was ordered from Sigma (Monoclonal ANTI-FLAG M2, F3165-1MG). The horseradish-peroxidase-coupled Streptavidin was ordered from Abcam (ab7403). The secondary antibodies were ordered from BioRad (Goat Anti-Rabbit IgG (H+L)-HRP Conjugate #172-1019, Goat Anti-Mouse IgG (H + L)-HRP Conjugate #172-1011). Antibodies were diluted in 5% (w/v) nonfat dry milk-TBS (Roth T145.2) with the following dilutions: anti-Mia40 1:5000, anti-Erv1 1:1000 anti-Ilv5 1:5000, anti-Tom70 1:2500, anti-Mam33 1:500, anti-Mrp17 1:500, anti-Mrpl40 1:500, anti-Mrpl36 1:500, anti-Mrp10 1:100, anti-Tim44 1:500, anti-FLAG 1:2000, anti-Rabbit/Mouse 1:10,000. Streptavidin was diluted 1:3333 in 3% BSA-TBS (Roth 8076.2).

## Analysis of mRNA levels by qRT-PCR

For total RNA extraction, yeast strains were cultivated in synthetic media to mid-log phase. 4 $OD_{600}$ of cells were harvested for RNA extraction using the RNeasy Mini Kit (Qiagen) in conjunction with the RNase-Free DNase Set (Qiagen) according to the manufacturer's instructions. Yield and purity of the obtained RNA were determined with a Spectrophotometer/Fluorometer DS-11 FX+ (DeNovix). In total, 500 ng RNA were reverse transcribed into cDNA using the qScript cDNA Synthesis Kit (Quanta Biosciences) according to the manufacturer's instructions. To measure relative mRNA levels, the iTaq Universal SYBR Green Supermix (BioRad) was used with 2 µl of a 1:10 dilution of cDNA sample. Measurements were performed in technical triplicate with the CFX96 Touch Real-Time PCR Detection System (BioRad). Calculations of the relative mRNA expressions were conducted following the 2-ΔΔCt method (Livak and Schmittgen, 2001). For normalization, the housekeeping gene *ACT1* was used due to its stability. The following primers were used: ACT1 fwd: AGAGTTGCCCCAGAAGAACACC, ACT1 rev: CGACGTGAG-TAACACCATCACC, TIM44 for: TTATTCTCCACCTCTACGACC, TIM44 rev: TCAGACTCGCCTAACTTTCC.

## Analysis of rRNA levels by qRT-PCR

Yeast strains were cultivated in SLac media, and expression was induced by 0.5% galactose. After 5 h of induction 50 $OD_{600}$ of cells were harvested by centrifugation (5000 $\times g$, 5 min). After a washing step, cell lysates were prepared in lysis buffer (10 mM Tris pH 7.4, 150 mM NaCl, 0.5% Triton, 1 mM PMSF, 1× Protease inhibitor, 1 U/µl RNAse inhibitor) using a FastPrep-24 5 G homogenizer (MP Biomedicals, Heidelberg, Germany) with 3 cycles of 20 s, speed 6.0 m/s, 120 s breaks, glass beads. Lysates were cleared by centrifugation (12,000 $\times g$, 5 min, 4 °C). Lysed samples were added to washed GFP-Trap-magnetic beads and incubated 1 h tumbling at 4 °C. The beads were washed 2× with wash buffer I (150 mM NaCl, 50 mM Tris pH 7.5, 5% glycerol, 0.05% Tx-100, RNAse inhibitor) and 1× with wash buffer II (150 mM NaCl, 50 mM Tris pH 7.5, 5% glycerol, RNAse inhibitor). For Elution beads were incubated with the lysis buffer of the RNeasy Mini Kit (Qiagen) for 10 min at 37 °C. RNA was extracted using the RNeasy Mini Kit (Qiagen) in conjunction with the RNase-Free DNase Set (Qiagen) according to the manufacturer's instructions. Yield and purity of the obtained RNA was determined with a Spectrophotometer/Fluorometer DS-11 FX+ (DeNovix). To measure relative rRNA levels, the Luna Universal One-Step RT-qPCR (NEB) was used with 20 ng of RNA sample. Measurements were performed in technical triplicate with the CFX96 Touch Real-Time PCR Detection System (BioRad). Calculations of the relative rRNA expressions were conducted following the 2-ΔΔCt method (Livak and Schmittgen, 2001). The Following Primers were used: 35S rRNA fwd: CGATATCAAACG-TACCATTCCG, 35S rRNA rev: TCCCACCTATTCCCTCTTGC, 15S rRNA fwd: GTTAATCATAATGGTTTAAAGGATCCGTAG AATG, 15S rRNA rev: GGTATCGAATCCGTTTCGCTACT CTAG.

## Isolation of mitochondria

For the isolation of mitochondria, cells were grown in rich or selective galactose media to mid-log phase. Cultures for CRISPRi samples were additionally treated with ATc (960 ng/ml final) for 18 h. For APEX labeling, cultures were grown in synthetic media, diluted into synthetic media without biotin and harvested after 18 h. Cells were harvested (2800 $\times g$, 5 min, RT) in the exponential phase. After a washing step, cells were treated for 10 min with 2 ml per g wet weight MP1 buffer (100 mM DTT, 10 mM Tris pH unadjusted) at 30 °C. After washing with 1.2 M sorbitol, yeast cells were resuspended in 6.7 ml per g wet weight MP2 buffer (20 mM KPi buffer pH 7.4, 1.2 M sorbitol,

3 mg per g wet weight zymolyase 20 T from Seikagaku Biobusiness) and incubated for 1 h at 30 °C. Spheroplasts were collected via centrifugation at 4 °C and resuspended in ice-cold homogenization buffer (13.4 ml/g wet weight) (10 mM Tris pH 7.4, 1 mM EDTA pH 8, 0.2% fatty acid-free bovine serum albumin (BSA), 1 mM PMSF, 0.6 M sorbitol). Spheroplasts were disrupted by 12 strokes with a cooled glass Potter. Cell debris was removed via centrifugation at $1900 \times g$ for 5 min. The supernatant was centrifuged for 12 min at $17,700 \times g$ for 10 min to collect mitochondria. Mitochondria were resuspended in 1 ml of ice-cold SH buffer (0.6 M sorbitol, 20 mM HEPES pH 7.4). The mitochondria were diluted to a protein concentration of 10 mg/ml.

## Protein import into mitochondria

The TNT® Quick Coupled Transcription/Translation Kit from Promega was used for the synthesis of $^{35}$S-methionine-labeled proteins in reticulocyte lysate. 50 µg mitochondria were taken in import buffer (500 mM sorbitol, 50 mM HEPES pH 7.4, 80 mM KCl, 10 mM Mg(OAc)$_2$ and 2 mM KH$_2$PO$_4$), 2 mM ATP and 2 mM NADH. Indicated concentrations of carbonyl cyanide 3-chlorophenylhydrazone (CCCP) were added into the reaction mix to disturb the membrane potential. VAO (55.6 µg/mL valinomycin, 440 µg/mL antimycin A, 850 µg/mL oligomycin in ethanol (>99.8%, p.a.) was added to dissipate the membrane potential. For ATP depletion, 40 U/ml apyrase were added instead of ATP and NADH. Samples were incubated for 10 min at 30 °C unless indicated otherwise. The import reaction was started by the addition of 1% (v/v) reticulocyte lysate. Samples were taken after the indicated time points, and the reaction was stopped by a 1:10 dilution in ice-cold SH buffer supplemented with 100 µg/ml proteinase K. The samples were incubated on ice for 30 min to remove precursors which were not imported. The protease treatment was stopped by the addition of 2 mM PMSF. The samples were centrifuged 15 min at $30,000 \times g$ and 4 °C. The mitochondria were washed with 500 µl SH/KCl-buffer (0.6 M sorbitol, 20 mM HEPES/KOH pH 7.4, 150 mM KCl) and 2 mM PMSF. The mitochondria were reisolated by centrifugation for 15 min at $30,000 \times g$ and 4 °C, resuspended in sample buffer and resolved via SDS-PAGE.

## Radioactive in organello labeling of mitochondrial translation products

Isolated mitochondria (100 µg) were incubated in 1.5× in organello translation buffer (ioTL buffer, 20 mM HEPES/KOH pH 7.4, 15 mM KPi, 0.6 M sorbitol, 150 mM KCl, 12.66 mM MgSO$_4$, 12 µg/ml of all amino acid except methionine, 7.5 mM phosphoenolpyruvate, 6 mM ATP, 0.75 mM GTP, 5 mM α-ketoglutarate, 10 µg/ml pyruvate kinase) together with 1 µl $^{35}$S-methionine (of a 11 µCi solution) and incubated at 30 °C for 10 min shaking at 600 rpm. Incorporation of radioactive methionine was quenched by the addition of 8 mM cold methionine and the reaction was stopped by the addition of 1 ml ice-cold SH buffer. Mitochondria were pelleted by centrifugation ($20,000 \times g$, 10 min, 4 °C), resuspended in loading buffer and analyzed by SDS-PAGE and autoradiography.

## Proteinase K digest

In total, 100 µg isolated mitochondria are taken in SH buffer with or without 0.2 mg/ml proteinase K. Samples were incubated on ice for 30 min. To stop the digest 1 ml SH buffer with 2 mM PMSF was added. To reisolate mitochondria samples were centrifuged 10 min at $25,000 \times g$ at 4 °C. Pellet was resuspended in 50 µl loading buffer and boiled for 5 min at 96 °C. Samples were analyzed by SDS-PAGE.

## APEX labeling and affinity purification on streptavidin beads

In all, 100 µg of isolated mitochondria were taken in SH buffer (0.6 M sorbitol, 20 mM HEPES pH 7.4) and 100 µM biotin-phenol. Samples were incubated for 30 min at 30 °C. To start the labeling reaction, 1 mM H$_2$O$_2$ was added. After 2 or 5 min the reaction was stopped by adding an equal amount of ice-cold quencher solution (10 mM sodium ascorbate, 10 mM sodium azide, 5 mM Trolox in SH buffer) to the sample which was put on ice. Mitochondria were reisolated by centrifugation for 10 min at $25,000 \times g$ at 4 °C. Pellets were washed with ice-cold quencher solution and again centrifuged for 10 min at $25,000 \times g$ at 4 °C. Pellets were resuspended in loading buffer, boiled for 5 min, and analyzed by SDS-PAGE.

For Streptavidin pulldown, mitochondria were washed in quencher solution, and the samples were divided into two samples (Total and IP). Both were centrifuged for 10 min at $25,000 \times g$ at 4 °C. The pellet of the Total sample was resolved in loading buffer and boiled for 5 min. To lyse mitochondria, the IP sample was resuspended in RIPA buffer (20 mM NaPi pH 7.4, 150 mM NaCl, 1% NP-40 (IGEPAL), 0.5% (w/v) deoxycholate, 0.1% SDS, 1× Protease Inhibitor (cOmplete Tablets, Roche, 04 693 159 001) and incubated 30 min at 4 °C on a rotator. Lysed IP samples were added onto equilibrated Streptavidin beads (Pierce™ Streptavidin magnetic beads, Thermo Scientific, 815-968-0747) and incubated for 1 h at 4 °C on a rotator. Beads were separated on a magnetic rack and washed two times with RIPA buffer and two times with TBS-T (1× TBS, 0.1% Tween). Samples were eluted by the addition of 30 µl loading buffer and boiling for 5 min. Samples were analyzed by SDS-PAGE.

## Sample preparation and mass spectrometric identification of proteins

For the co-immunoprecipitation of GFP-Mrp17 and mass spectrometry cells were grown in SLac media and expression was induced by 1.25% galactose. For the co-immunoprecipitation of Apj1-GFP, Pwp2-GFP and Utp18-GFP and mass spectrometry cells were grown in SLac media and expression of b$_2$-DHFR (clogger) was induced by 0.5% galactose. After 5 h of induction 20 OD$_{600}$ of cells were harvested by centrifugation ($5000 \times g$, 5 min). Cell lysates were prepared in lysis buffer (10 mM Tris pH 7.4, 150 mM NaCl, 0.5 mM EDTA, 0.5% Triton, 1 mM PMSF, 1× Protease inhibitor) using a FastPrep-24 5 G homogenizer (MP Biomedicals, Heidelberg, Germany) with three cycles of 20 s, speed 6.0 m/s, 120 s breaks, glass beads. Lysates were cleared by centrifugation ($12,000 \times g$, 5 min, 4 °C). Washed GFP-Trap-magnetic beads were added to lysed samples and incubated 1 h tumbling at 4 °C. The beads were washed 2× with wash buffer I (150 mM NaCl, 50 mM Tris pH 7.5, 5% glycerol, 0.05% Tx-100) and 2× with wash buffer II (150 mM NaCl, 50 mM Tris pH 7.5, 5% glycerol).

For Streptavidin pulldown, cells were grown in galactose-containing synthetic media and starved for biotin for 18 h before

mitochondria were isolated. For APEX labeling 500 µg isolated mitochondria were resuspended in SH buffer (20 mM HEPES pH 7.4, 0.6 M sorbitol) and 100 µM biotin-phenol. After incubation at 30 °C for 30 min, 1 mM $H_2O_2$ was added. After 5 min, the labeling reaction was stopped by the addition of an equal amount of ice-cold quencher solution (10 mM sodium ascorbate, 10 mM sodium azide, 5 mM Trolox in SH buffer), and samples were put on ice. After centrifugation (10 min at $25,000 \times g$, 4 °C), the pellets were washed with quencher solution. After another centrifugation step (10 min, $25,000 \times g$, 4 °C) pellets were resuspended in ice-cold RIPA buffer (20 mM NaPi pH 7.4, 150 mM NaCl, 1% NP-40, 0.5% deoxycholate, 0.1% SDS, 1x Protease inhibitor). To lyse mitochondria, samples were incubated 30 min at 4 °C. Cleared lysates ($20,000 \times g$, 10 min, 4 °C) were incubated with activated Streptavidin beads (Pierce™ Streptavidin magnetic beads, Thermo Scientific, 815-968-0747) and tumbled end-over-end for 1 h at 4 °C. The supernatant was discarded, and Streptavidin beads were washed three times with RIPA buffer and two times with 20 mM Tris.

Peptides were digested on-bead with Elution buffer I (2 M urea, 50 mM Tris-HCl, pH 7.5, 1 mM DTT, 5 ng/µl trypsin (Promega, #V5111)) for 1 h at room temperature. Eluted peptides were transferred to a fresh tube. For a second elution step, 50 µl Elution buffer II (2 M urea, 50 mM Tris pH 7.5, 5 mM chloroacetamide, 5 ng/µl trypsin) were again added to the beads. After 30 min at room temperature, the supernatant was combined with the first elution. Samples were incubated overnight in the dark at 37 °C. Peptides were acidified to pH <2 with trifluoroacetic acid and desalted on homemade StageTips containing Empore $C_{18}$ disks (Rappsilber et al, 2007). C18 stage tips were activated with 100 µl methanol, 100 µl buffer B (0.1% formic acid, 80% acetonitrile) and twice with 100 µl buffer A (0.1% formic acid). The acidified peptides were added onto the stage tips and washed with 100 µl buffer A. Peptides were eluted with 40 µl buffer B and dried-down in a speed vac and resolubilized in 9 µl buffer A (0.1% formic acid in MS grade water) and 1 µl buffer A* (2% acetonitrile, 0.1% tri-flouracetic acid in MS grade water). Peptides were separated using an Easy-nLC 1200 system (Thermo Scientific) coupled to a Q Exactive HF mass spectrometer via a Nanospray-Flex ion source. The analytical column (50 cm, 75 µm inner diameter (New-Objective) packed in-house with C18 resin ReproSilPur 120, 1.9 µm diameter Dr. Maisch) was operated at a constant flow rate of 250 nl/min. Gradients of 90 min were used to elute peptides (Solvent A: aqueous 0.1% formic acid; Solvent B: 80% acetonitrile, 0.1% formic acid). MS spectra with a mass range of 300–1.650 m/z were acquired in profile mode using a resolution of 60,000 [maximum fill time of 20 ms or a maximum of 3e6 ions (automatic gain control, AGC)]. Fragmentation was triggered for the top 15 peaks with charge 2–8 on the MS scan (data-dependent acquisition) with a 20 s dynamic exclusion window (normalized collision energy was 28). Precursors were isolated with a 1.4 $m/z$ window, and MS/MS spectra were acquired in profile mode with a resolution of 15,000 (maximum fill time of 80 ms, AGC target of 2e4 ions).

## Analysis of mass spectrometry data

Peptide and protein identification and quantification was done using the MaxQuant software (version 2.0.1.0) (Cox and Mann, 2008; Cox et al, 2011; Tyanova et al, 2016) and a *S. cerevisiae* proteome database obtained from Uniprot. Protein N-terminal acetylation and Met oxidation were specified as variable modifications and Cys carbamidomethylation as fixed modification. The "Requantify" and "Second Peptides" options were deactivated. False discovery rate was set at 1% for peptides, proteins and sites, minimal peptide length was seven amino acids. For label-free data, the LFQ normalization algorithm and second peptides was enabled. Match between run was applied within each group of replicates.

The protein groups identified in each mass spectrometry dataset were processed and analyzed in parallel using the R programming language (R Core Team, 2018, version 4.2.1). First, the MaxQuant output was filtered to remove contaminants, reverse hits, proteins identified by site only as well as proteins that were identified in less than three replicates ($N = 4$) of every condition. This resulted in 1744 (GFP-pulldown), 1341 (Streptavidin pulldowns) robustly identified protein groups whose label-free quantification (LFQ) intensities were log2-transformed subsequently. Lastly, in the GFP-Mrp17 Pull-down missing values in the Control samples were imputed by sampling $n = 4$ values from a normal distribution and using them whenever there are no valid values in a quadruplicate of a condition. Different for each dataset, the mean of this normal distribution corresponds to the 1% percentile of LFQ intensities, and its standard deviation is determined as the median of LFQ intensity sample standard deviations calculated within and then averaged over each quadruplicate.

Principal component analysis was carried out for each dataset using the processed and standardized LFQ intensities of those protein groups with an ANOVA F-statistic $P$ value $< 0.05$ between all replicate groups to filter for proteins with a discernable degree of variance between conditions.

For the calculation of fold changes and $p$ values, the limma package within the R programming language was used (Ritchie et al, 2015). Protein groups were statistically analyzed using pairwise, two-sided Welch's $t$ tests on the processed LFQ intensities between the replicates of respective control and treatment conditions. Only proteins with valid values in the bait pulldown were considered for comparison. Log2 fold changes were derived as the tested difference of means and the resulting $P$ values were adjusted for multiple testing using the Benjamini-Hochberg procedure (Benjamini and Hochberg, 1995).

For gene ontology (GO) enrichment analysis, proteins with smaller than −0.5 log2 fold change were used as target set and analyzed by using the GOrilla tool (http://cbl-gorilla.cs.technion.ac.il/) with all quantified proteins as the background set.

## Fluorescence microscopy

For microscopy, cells were grown to mid-log phase. In all, 1 OD was harvested via centrifugation and the cell pellets were resuspended in 30 µl of $H_2O$. In total, 3 µl were pipetted onto a glass slide and covered with a cover slip. Manual microscopy was performed using a Leica Dmi8 Thunder Imager. Images were acquired using an HC PL APO100×/1.44 Oil UV objective with Immersion Oil Type A 518 F. For excitation of GFP 475 nm, mNeonGreen 510 nm, mScarletI 575 nm, Tomato 555 nm was used. All images were taken as Z-stacks. Image analysis was done with the LAS X software, and further processing of images was performed in Fiji/ImageJ.

## Staining with MitoTracker

Cells were grown in liquid SGal media to mid-log phase. 1 OD was harvested via centrifugation (1 min, $16,000 \times g$) and the cell pellets

were washed with 1× PBS. Cells were resuspended in MitoTracker Red CMXRos (200 nM in 1× PBS) and incubated for 10 min at room temperature in the dark. Cells were washed 3× with 1× PBS. Cells were centrifuged (1 min, 16,000 ×$g$), and cell pellet was resuspended in 15 µl 1× PBS. Cells were imaged as described above. For excitation, 575 nm was used.

## Fluorescence intensity measurements

To measure the fluorescence intensity of cells stained with MitoTracker, the CLARIOstar fluorescence plate reader by BMG Labtech was used. To this end, 4 OD$_{600}$ of yeast cells were harvested and prepared as described above. In all, 100 µl of the cell suspension were transferred to flat-bottomed black 96-well imaging plates (BD Falcon, Heidelberg, Germany) in technical triplicate. Cells were sedimented by gentle spinning (30× $g$, 5 min, RT), and fluorescence (excitation 575 nm, emission 599 nm) was measured. Unstained cells were used for background subtraction of autofluorescence.

## Data availability

The mass spectrometry proteomics data (see also Datasets EV1, 4 and 5) have been deposited to the ProteomeXchange Consortium via the PRIDE (Perez-Riverol et al, 2019) partner repository with the dataset identifiers PXD057245 for Dataset 1, PXD062600 for Dataset 4 and PXD054381 for Dataset 5. No specific coding was used for this study. The R scripts for the analysis of proteomics data is available from the authors upon request.

The source data of this paper are collected in the following database record: biostudies:S-SCDT-10_1038-S44318-025-00486-1.

## Peer review information

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

## Acknowledgements

The authors thank Vera Nehr for assistance, Christian Koch for help with the proteomics data analysis, Lukas Nutz for cloning split-GFP plasmids, Yury Bykov and Katja G Hansen for critical reading of the manuscript, Dejana Mocranjac for the Tim44 antibody, Jean-Paul Di Rago for the *Δatp6* mutant strain, and Alice Ting for a plasmid containing the APEX2 sequence. The authors are grateful for financial support from the Deutsche Forschungsgemeinschaft (HE2803/9-2 to JMH), the European Research Council (MitoCyto 101052639 to JMH) and the Research Initiative of Rheinland-Pfalz (BioComp to JMH).

## Author contributions

**Tamara Flohr**: Conceptualization; Resources; Data curation; Formal analysis; Validation; Investigation; Methodology; Writing—original draft. **Markus Räschle**: Conceptualization; Resources; Validation; Investigation; Writing—review and editing. **Johannes M Herrmann**: Conceptualization; Data curation; Supervision; Funding acquisition; Investigation; Visualization; Writing—review and editing.

Source data underlying figure panels in this paper may have individual authorship assigned. Where available, figure panel/source data authorship is listed in the following database record: biostudies:S-SCDT-10_1038-S44318-025-00486-1.

## Funding

## Disclosure and competing interests statement

The authors declare no competing interests.

# Expanded View Figures

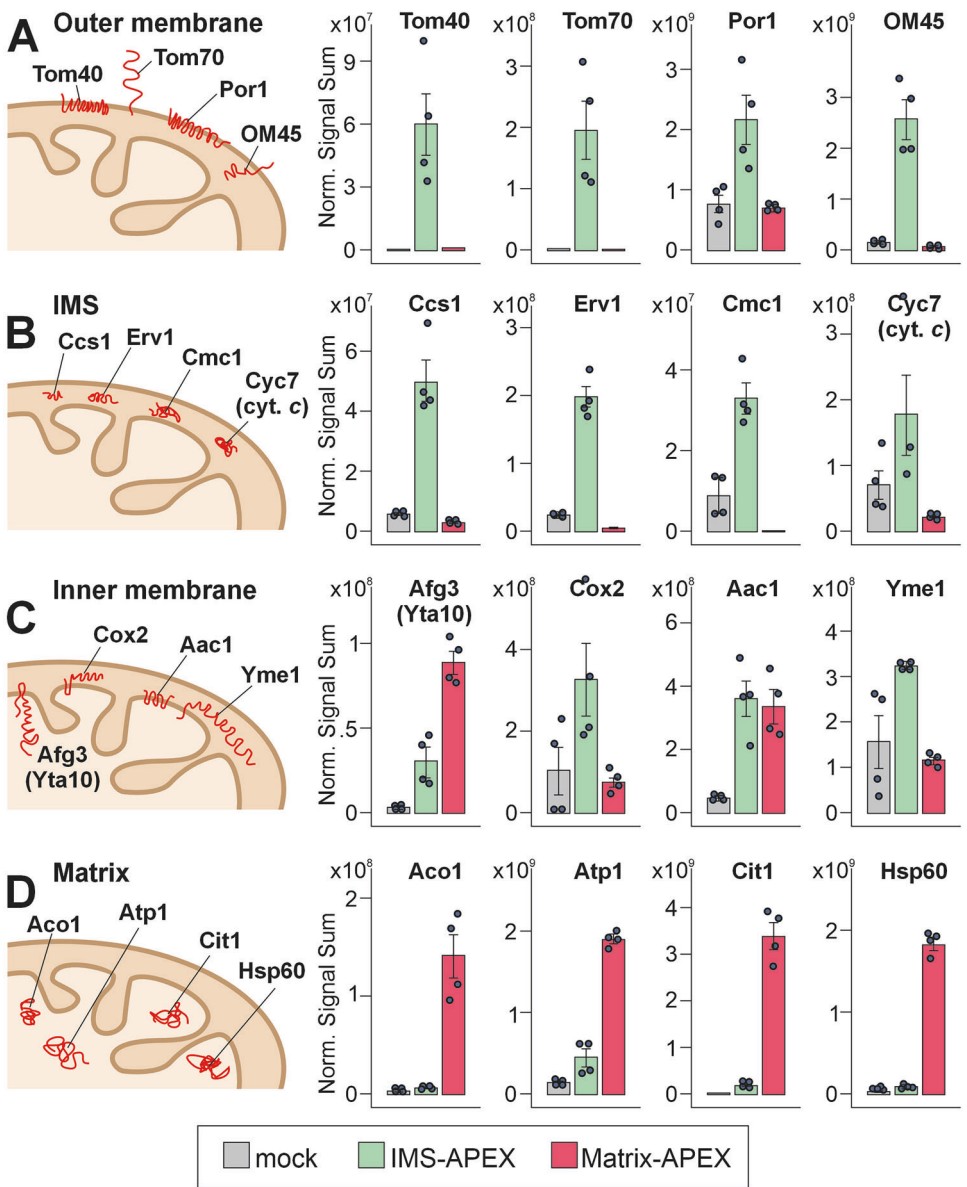

**Figure EV1. APEX-mediated biotinylation patterns reveal intramitochondrial distribution of proteins.**

(A–D) Protein levels were detected by mass spectrometry in the streptavidin pulldowns from lysed isolated mitochondria, derived from cells expressing no APEX (gray), or IMS-APEX (green) or matrix-APEX (red). Shown are the normalized signal sums of mitochondrial proteins of the four different mitochondrial subcompartments. Plotted were mean values and standard deviations from four replicates. The intramitochondrial localizations of the proteins are sketched on the left. See Dataset EV1 for details.

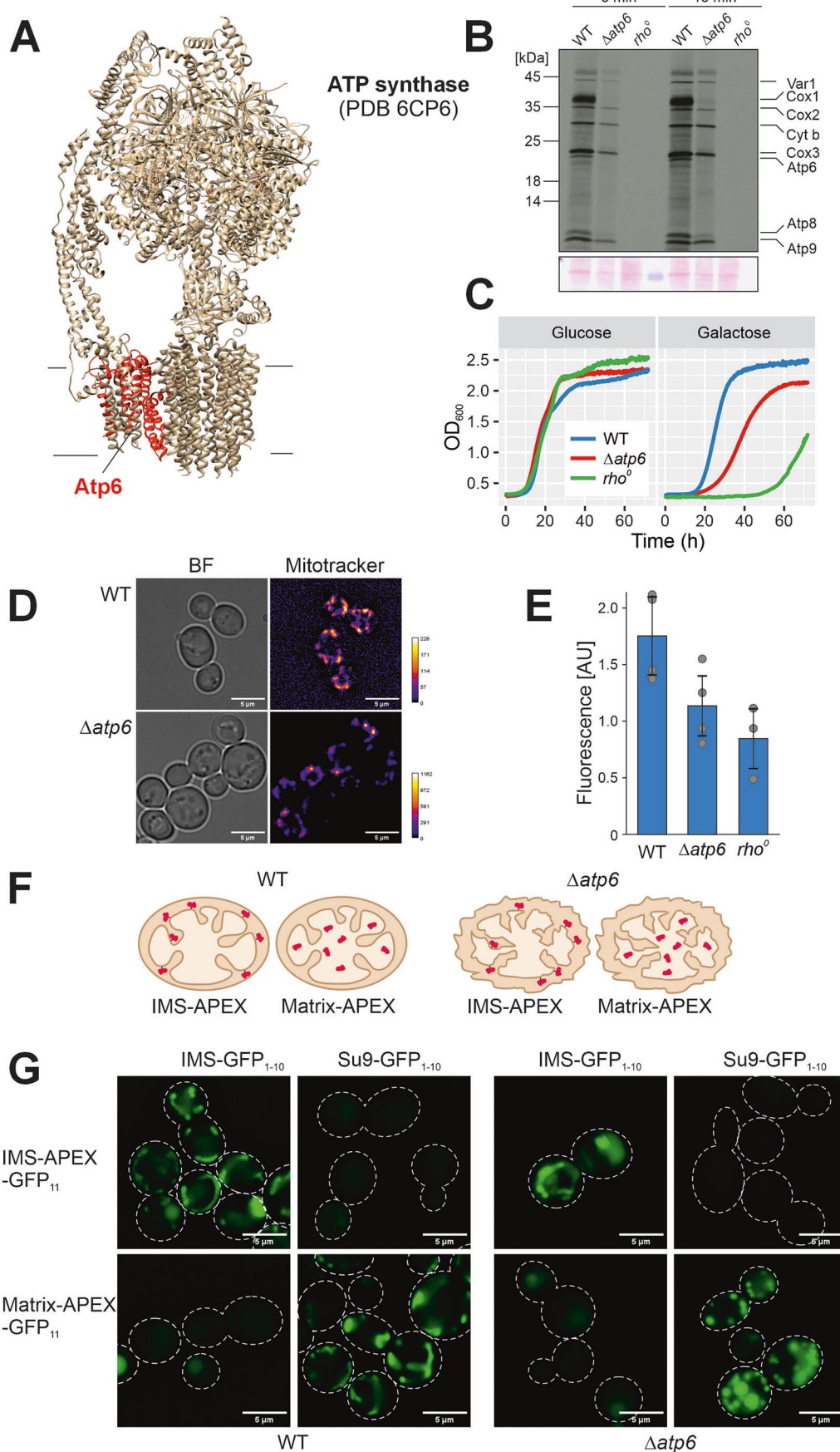

**Figure EV2.   A mutant lacking the mitochondrially encoded *ATP6* gene serves as example for mitochondrial dysfunction.**

(A) Structure of the ATPase complex of baker's yeast (Srivastava et al, 2018). Atp6 (shown in red) is a subunit of the $F_o$ part that plays a crucial role in proton pumping. (B) Mitochondria were isolated from the indicated strains and incubated in *in organello* translation buffer in the presence of $^{35}$S-methionine for 5 or 15 min. Radiolabeled translation products were separated by SDS-PAGE and visualized by autoradiography. (C) Growth of the indicated strains in synthetic medium containing the indicated carbon sources was recorded continuously and plotted. Shown are mean values of three technical replicates. (D) Wild-type and *Δatp6* cells were grown in galactose medium. Mitochondria were stained with the membrane potential-dependent dye mitotracker CMXRos. Intensity values were calculated to indicate the degree of mitochondrial energization in the two strains. (E) The fluorescence intensities of the mitotracker-stained mitochondria were quantified as a measure of the mitochondrial membrane potential in the strains. Shown are mean values and standard deviations from four biological replicates. (F) Schematic representation of intramitochondrial distribution of the APEX fusion proteins in the different strains used in this study. (G) The IMS-APEX and matrix-APEX reporters were fused to the eleventh beta sheet of GFP (GFP$_{11}$) and coexpressed with IMS- and matrix-targeted fragments of the first 10 beta sheets of GFP (GFP$_{1-10}$). Signals indicate colocalization with these split-GFP reporters (Cabantous and Waldo, 2006). Source data are available online for this figure.

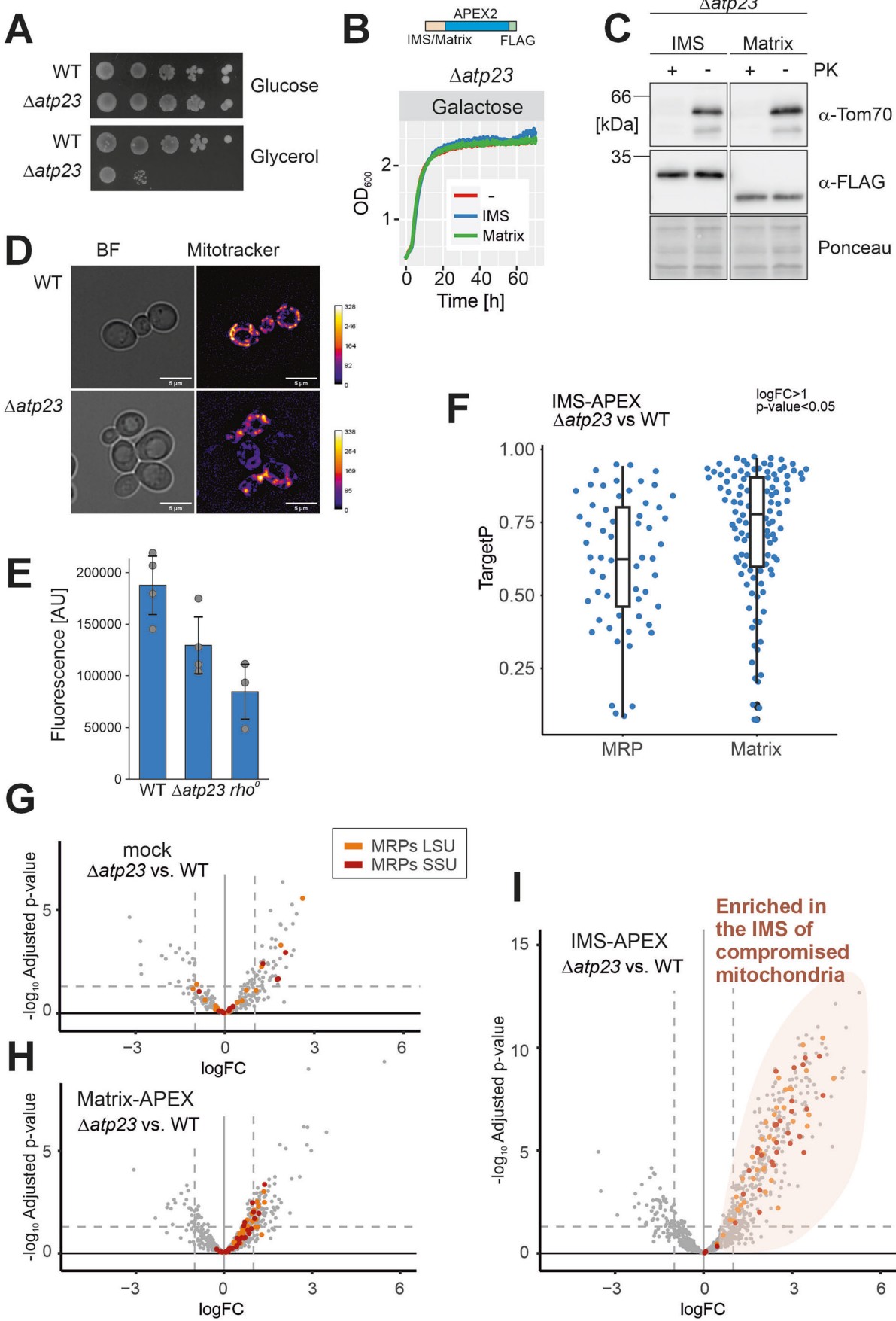

**Figure EV3. A nuclear model for mitochondrial dysfunction phenocopies the situation in Δ*atp6* cells.**

(A) Wild-type (WT) and Δ*atp23* cells were grown to log phase in galactose-containing medium before tenfold serial dilutions were dropped onto medium containing glucose or glycerol as carbon sources. (B) The growth of Δ*atp23* cells expressing the indicated APEX reporters was measured continuously. (C) Mitochondria were isolated from wild-type and Δ*atp23* cells and incubated with or without proteinase K (PK) to remove non-imported proteins. The presence of the surface protein Tom70 and of the FLAG-tagged APEX fusion proteins was verified by Western blotting. (D) Wild-type and Δ*atp23* cells were grown in galactose medium. Mitochondria were stained with the membrane potential-dependent dye mitotracker CMXRos. Intensity values were calculated to indicate the degree of mitochondrial energization in the two strains. (E) The fluorescence intensities of the mitotracker-stained mitochondria were quantified as a measure of the mitochondrial membrane potential in the strains. Shown are mean values and standard deviations from four biological replicates. (F) Comparison of the TargetP values of IMS-located MRPs in comparison to other matrix proteins that are enriched in the IMS of Δ*atp23* mitochondria. Shown are proteins that are significantly enriched logFC > 1, *P* value < 0.05. (G–I) Volcano plots showing the relative distribution of proteins purified with streptavidin beads in mitochondria from wild-type and Δ*atp23* cells. The $\log_2$ fold change (logFC) values were calculated from $n = 4$ samples. Proteins of the small subunit of the mitoribosome are colored in red and of the large subunit in orange. Significantly enriched proteins are indicated by the dashed lines (logFC > 1, *P* value < 0.05). Please note the strong accumulation of MRPs (shown in color) in the IMS of Δ*atp23* cells, that is reminiscent to the situation in Δ*atp6* cells (Fig. 2F). Source data are available online for this figure.

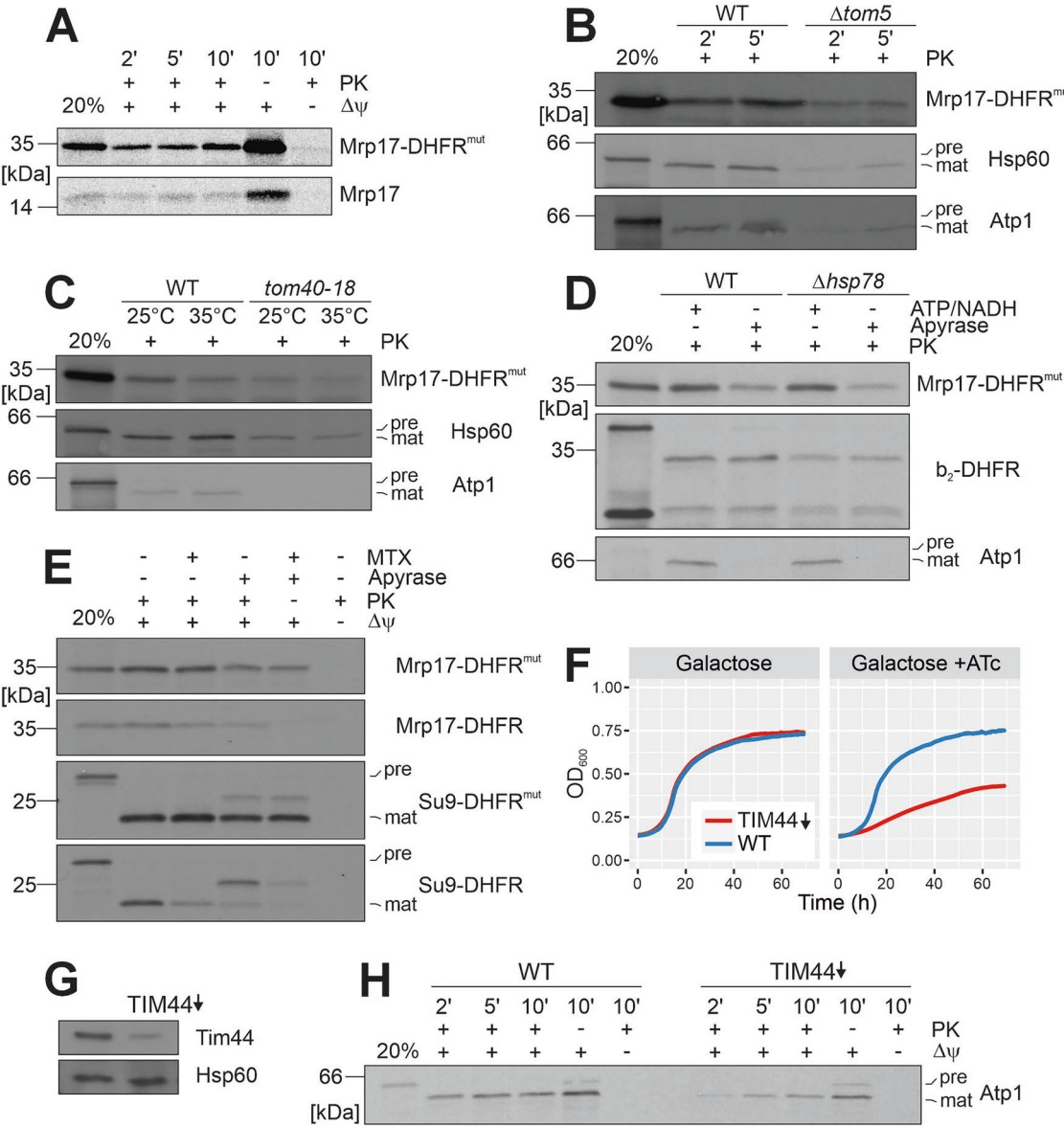

**Figure EV4. Mrp17 is imported into mitochondria independent on the function of the import motor.**

(A) Radiolabeled Mrp17-DHFR^mut and Mrp17 were incubated with isolated wild-type mitochondria for the times indicated. Mrp17 has only four methionine residues (including the N-terminal one); the fusion to the permanently unfolded DHFR domain (Vestweber and Schatz, 1988) increased the number of methionine residues leading to considerably enhanced radiolabeled signals of the protein (Bykov et al, 2022). The membrane potential (Δψ) was dissipated where indicated. Mitochondria were reisolated and incubated with proteinase K (PK). 20% of the radiolabeled protein used per import lane was loaded for control. (B–D) Radiolabeled proteins were incubated with mitochondria isolated from the indicated strains. In case of the temperature-sensitive *tom40-18* cells and the corresponding wild type (C), mitochondria were pretreated for 10 min at 35 °C before the experiment. (E) The indicated proteins were synthesized in reticulocyte extract in the presence or absence of 5 μM methotrexate (MTX) to stabilize the folding of DHFR, before import experiments into wild-type mitochondria were carried out for 5 min. (F) Wild-type cells containing the TIM44-CRISPRi plasmid were grown in the absence or presence of 960 ng/μl anhydrotetracycline (ATc) at 30 °C. Cells were preincubated for 6 h with ATc before the growth curve measurement. Shown are mean values of three technical replicates. (G) The expression of Tim44 was repressed by induction of the Tim44-CRISPRi plasmid for 16 h before mitochondria were isolated and analyzed by Western blotting. (H) Mitochondria were isolated from cells in which the expression of Tim44 had been repressed for 16 h. Radiolabeled Atp1 was imported into these mitochondria for the times indicated. Source data are available online for this figure.

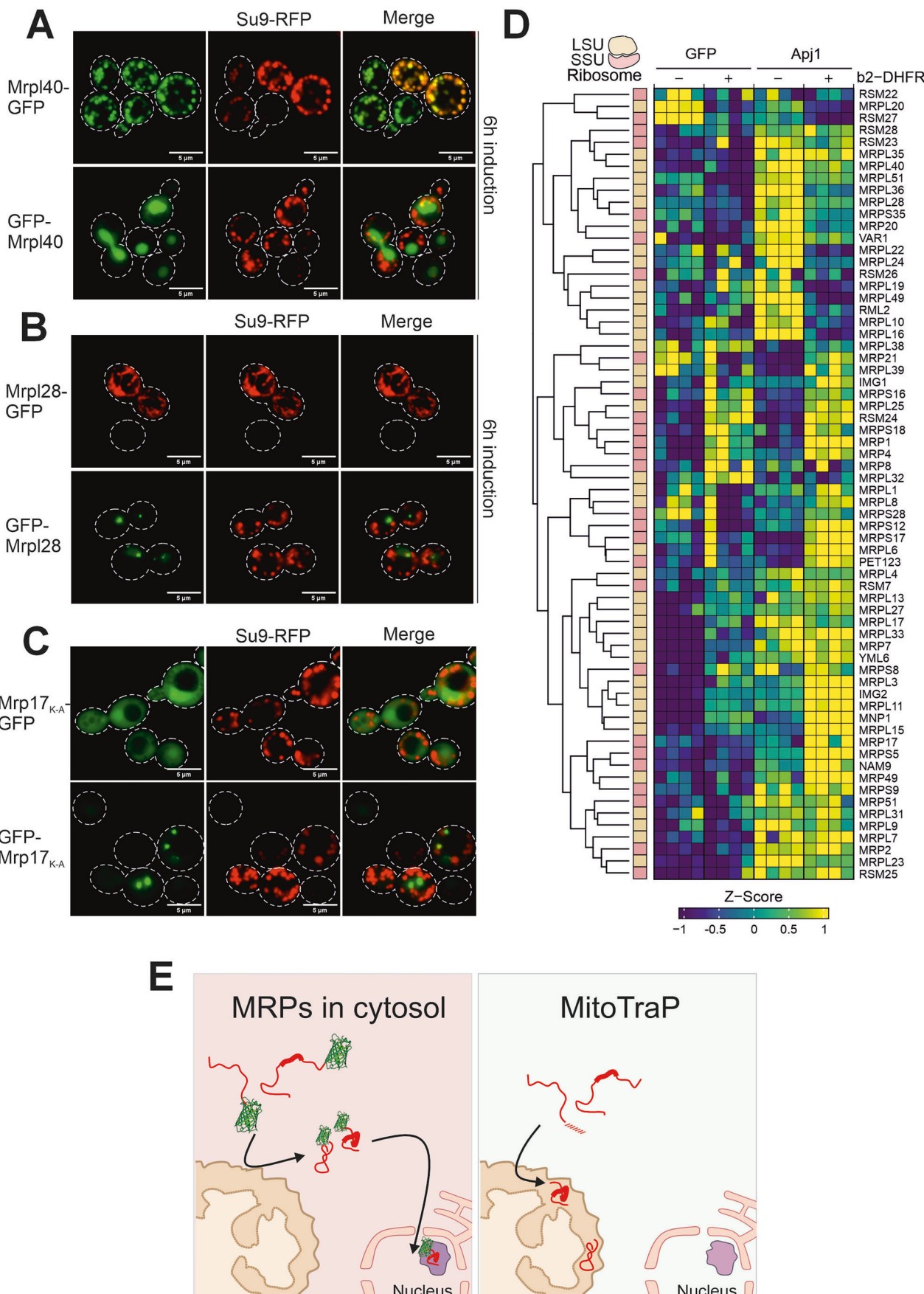

◀ **Figure EV5. The non-importable GFP-Mrp17 variant associates with components of the 90S pre-ribosome assembly complexes.**

(A–C) GFP-Mrpl40, GFP-Mrpl28 and GFP-Mrp17$_{K-A}$ were expressed and visualized by fluorescence microscopy 6 h after induction in galactose-containing medium. Mitochondria were stained by expression of Su9-RFP. (D) Heat map showing the intensities of MRPs there were co-isolated with GFP or Apj1-GFP in the presence or absence of the clogger b$_2$-DHFR. Proteins were filtered for small and large subunits of the mitochondrial ribosomes. Imputed LFQ intensities were z-score normalized across rows and subject to non-supervised hierarchical clustering. $n = 4$. (E) Mitochondrial dysfunction leads to the trapping of MRPs in the IMS. The accumulation of MRPs in the IMS might prevent their targeting to the nucleus and, thus, their interference with the assembly of ribosomes in the nucleolus.

