## [Peer Review File · The EMBO Journal]

Dysfunctional mitochondria trap proteins in the intermembrane space

Tamara Flohr, Markus Räschle, and Johannes Herrmann

Corresponding author(s): Johannes Herrmann (hannes.herrmann@biologie.uni-kl.de)

Review Timeline:

Submission Date:	28th Oct 24
Editorial Decision:	28th Nov 24
Revision Received:	15th Apr 25
Editorial Decision:	13th May 25
Revision Received:	14th May 25
Accepted:	27th May 25

Editor: William Teale

Transaction Report:

Dear Hannes,

Thank you again for the submission of your manuscript entitled "Dysfunctional mitochondria trap proteins in the intermembrane space" and for your patience during the review process. We have now received the reports from the referees, which I copy below.

As you can see from their comments, none of the referees were convinced by your hypothesis that the import sequence of mitochondrial ribosome proteins is adapted by pressure from disrupted nucleolar ribosomal assembly. Importantly, referee #1 and referee #3 considered the data to be interesting, novel and robust. Referee #2 was, however, less convinced. All raise points that will require your attention before your manuscript can be published in The EMBO Journal.

Based on the overall interest expressed in the reports, I would like to invite you to address the comments of all referees in a revised version of the manuscript. I should add that it is The EMBO Journal policy to allow only a single major round of revision and that it is therefore important to resolve the main concerns at this stage. I believe the concerns of the referees are reasonable and addressable, but please contact me if you have any questions (I am available for a Zoom call at any time), need further input on the referee comments or if you anticipate any problems in addressing any of their points. Please, follow the instructions below when preparing your manuscript for resubmission.

I would also like to point out that as a matter of policy, competing manuscripts published during this period will not be taken into consideration in our assessment of the novelty presented by your study ("scooping" protection). We have extended this 'scooping protection policy' beyond the usual 3 month revision timeline to cover the period required for a full revision to address the essential experimental issues. Please contact me if you see a paper with related content published elsewhere to discuss the appropriate course of action.

Again, please contact me at any time during revision if you need any help or have further questions.

Thank you very much again for the opportunity to consider your work for publication. I look forward to your revision.

Best regards,

William

William Teale, Ph.D.
Editor
The EMBO Journal

When submitting your revised manuscript, please carefully review the instructions below and include the following items:

- 1) a .docx formatted version of the manuscript text (including legends for main figures, EV figures and tables). Please make sure that the changes are highlighted to be clearly visible.
- 2) individual production quality figure files as .eps, .tif, .jpg (one file per figure).
- 3) a .docx formatted letter INCLUDING the reviewers' reports and your detailed point-by-point response to their comments. As part of the EMBO Press transparent editorial process, the point-by-point response is part of the Review Process File (RPF), which will be published alongside your paper.
- 4) a complete author checklist, which you can download from our author guidelines ([https://wol-prod-cdn.literatumonline.com/pb-assets/embo-site/Author Checklist%20-%20EMBO%20J-1561436015657.xlsx](https://wol-prod-cdn.literatumonline.com/pb-assets/embo-site/Author%20Checklist%20-%20EMBO%20J-1561436015657.xlsx)). Please insert information in the checklist that is also reflected in the manuscript. The completed author checklist will also be part of the RPF.
- 5) Please note that all corresponding authors are required to supply an ORCID ID for their name upon submission of a revised manuscript.
- 6) We require a 'Data Availability' section after the Materials and Methods. Before submitting your revision, primary datasets produced in this study need to be deposited in an appropriate public database, and the accession numbers and database listed

under 'Data Availability'. Please remember to provide a reviewer password if the datasets are not yet public (see <https://www.embopress.org/page/journal/14602075/authorguide#datadeposition>). If no data deposition in external databases is needed for this paper, please then state in this section: This study includes no data deposited in external repositories. Note that the Data Availability Section is restricted to new primary data that are part of this study.

Note - All links should resolve to a page where the data can be accessed.

8) For data quantification: please specify the name of the statistical test used to generate error bars and P values, the number (n) of independent experiments (specify technical or biological replicates) underlying each data point and the test used to calculate p-values in each figure legend. The figure legends should contain a basic description of n, P and the test applied. Graphs must include a description of the bars and the error bars (s.d., s.e.m.).

9) We would also encourage you to include the source data for figure panels that show essential data. Numerical data can be provided as individual .xls or .csv files (including a tab describing the data). For 'blots' or microscopy, uncropped images should be submitted (using a zip archive or a single pdf per main figure if multiple images need to be supplied for one panel). Additional information on source data and instruction on how to label the files are available at .

10) We replaced Supplementary Information with Expanded View (EV) Figures and Tables that are collapsible/expandable online (see examples in <https://www.embopress.org/doi/10.15252/embj.201695874>). A maximum of 5 EV Figures can be typeset. EV Figures should be cited as 'Figure EV1, Figure EV2" etc. in the text and their respective legends should be included in the main text after the legends of regular figures.

12) Our journal encourages inclusion of *data citations in the reference list* to directly cite datasets that were re-used and obtained from public databases. Data citations in the article text are distinct from normal bibliographical citations and should directly link to the database records from which the data can be accessed. In the main text, data citations are formatted as follows: "Data ref: Smith et al, 2001" or "Data ref: NCBI Sequence Read Archive PRJNA342805, 2017". In the Reference list, data citations must be labeled with "[DATASET]". A data reference must provide the database name, accession number/identifiers and a resolvable link to the landing page from which the data can be accessed at the end of the reference. Further instructions are available at .

13) In order to increase the reproducibility and reach of your work, The EMBO Journal includes a table of reagents that were used in the study. Please provide this along with your revisions.

Further instructions for preparing your revised manuscript:

When assembling figures, please refer to our figure preparation guideline in order to ensure proper formatting and readability in

print as well as on screen:

We realize that it is difficult to revise to a specific deadline. In the interest of protecting the conceptual advance provided by the work, we recommend a revision within 3 months (26th Feb 2025). Please discuss the revision progress ahead of this time with the editor if you require more time to complete the revisions. Use the link below to submit your revision:

Referee #1:

In this interesting manuscript, the authors report the novel observation that precursors of mitochondrial matrix proteins can accumulate in the intermembrane space (IMS) of ATP-depleted mitochondria. Using a proximity labeling approach, they identified proteins of the mitochondrial ribosome in the IMS of yeast mitochondria lacking the mitochondrially encoded protein ATP6. This finding was supported by a series of convincing in vitro experiments with isolated mitochondria. While this study provides valuable insights into the dynamics of mitochondrial protein targeting, the extent, mechanism and functional consequences of precursor accumulation in the IMS remain to be fully elucidated. At present, the authors' interpretation that the observed accumulation of mitochondrial ribosomal proteins in the IMS serves as a protective mechanism rather than a mistargeting event would require further experimental support. Alternatively, the authors could tone down this conclusion, which would not take away from the interest in their findings.

Major points:

1. To assess the extent of the IMS accumulation, the authors could quantify the fraction of mitochondrial ribosomal proteins (e.g. MRP17) that accumulate in the IMS in defective mitochondria relative to the corresponding protein level under normal conditions in the mitochondrial matrix of yeast cells. In addition, it would be informative to determine how much mitochondrial ribosomal precursor protein would accumulate outside of mitochondria if IMS accumulation is prevented (e.g., by disrupting the membrane potential)?
2. The authors suggest that non-canonical targeting sequences of mitochondrial proteins may mediate targeting to the IMS. To support this argument, the authors could perform comparative experiments with two (or more) mitochondrial ribosomal proteins, one containing an internal targeting sequence and the other containing a conventional N-terminal targeting sequence.
3. The authors claim that non-imported mitochondrial ribosomal proteins interfere with cytosolic ribosome biogenesis. To experimentally validate this claim, the authors can quantify fully assembled ribosomes by performing a sucrose gradient fractionation experiment in yeast cells expressing a GFP tagged ribosomal protein.

Minor points:

1. Figure 1B and 5B, C: Labeling of the Scale bar is too small.

2. Figure 3C: The presence of Proteinase K is not indicated as in the other panels of the figure.

Referee #2:

Compromised energy-generating functions of mitochondria often lead to the accumulation of non-import mitochondrial proteins, which can be toxic to the cell. In this manuscript, Flohr et al report that mitochondrial ribosomal proteins (MRPs) are sequestered in the intermembrane space (IMS) rather than accumulating outside mitochondria when the mitochondrial ATP levels, which are generally required for protein import into the matrix, are lowered. This IMS sequestration may serve a mechanism to prevent mistargeting of MRPs to the nucleus, where they could interfere with ribosome assembly. The authors thus named this phenomenon "MitoTraP" (mitochondrial triage of precursor proteins), proposing it as a novel mechanism to reduce the toxic effects of non-imported mitochondrial proteins. The experiments were well-designed and carefully performed, and the results were analyzed properly. The findings are thought-provoking and the proposed model is worthy of consideration. Nevertheless, the idea of MitoTraP is, at least at the moment and as it stands, not convincing enough for its generality.

The major concern is that the authors did not rule out the possibility that the observed IMS accumulation of MRPs might merely reflect dead-end products of precursor proteins at the TIM23 complex level due to compromised motor function of mitochondrial Hsp70. The authors argued that "this IMS accumulation was not simply the result of an import block of matrix proteins as the mislocalization affected only a specific subgroup of proteins", but MRP's untypical targeting signals may render them sensitive to reduced import motor function, making their translocation across the IM more vulnerable compared to precursor proteins with canonical N-terminal mitochondrial targeting sequences.

The authors state that "almost the full complement of subunits of this assembly complex were co-isolated with GFP-Mrp17 (Fig. 5F)" (page 8). Then, what about other MRPs, such as Mrp40 and Mrp28, which also accumulate in the IMS? Further data addressing these MRPs would strengthen the study.

The authors also demonstrate that non-imported GFP-Mrp17 is targeted into the nucleus, where it interferes with the biogenesis of 80S ribosomes. Then, does overexpression of GFP-Mrp17 indeed interfere with cytosolic ribosome assembly and negatively affect cell growth? This prediction can be experimentally tested. In relation to this, is the GFP-Mrp17 accumulation in the nucleus comparable in *atp6Δ* cells or under CCCP treatment? Clarifying these points would provide additional insights into the proposed mechanism.

Other points.

Fig. 3. There is no explanation of DHFRmut and no description of the purpose of the experiments using the fusion proteins with DHFR and DHFRmut.

Appendix Fig. S2B. Why is labeling by Matrix-APEX weaker than by IMS-APEX in *atp6Δ* cells? This may raise concerns about the reliability of APEX labeling, as low labeling efficiency could allow proteins to escape detection. The authors had better address this potential limitation and emphasize the need of caution when interpreting the results.

Overall, the manuscript presents intriguing findings, but further experiments and clarifications are required to substantiate the generality and robustness of the proposed MitoTraP model.

Referee #3:

Dysfunctional mitochondria often exhibit impaired capacity for protein import, leading to the extramitochondrial accumulation of mitochondrial precursor proteins, which may cause proteotoxic effects within the cell. However, the mechanisms employed by eukaryotic cells to prevent accumulation of these proteins under conditions of mitochondrial stress are not well understood. In this manuscript, the authors propose that protein retention in the intermembrane space (IMS) of damaged mitochondria, a mechanism they termed MitoTraP, is exploited by the cell to sequester mitochondrial protein precursors and prevent their extramitochondrial accumulation.

It is demonstrated through an APEX labeling approach in yeast cells that a subset of proteins containing putative non-conventional mitochondrial targeting sequences, in particular mitoribosomal proteins (MRPs), are preferentially retained in the

IMS consequent to a mitochondrial defect induced by mutation of an FoF1 ATPase subunit. Using Mrp17 as a model substrate, it is demonstrated that these non-canonical proteins are transported into the IMS of compromised mitochondria independently of the presence of ATP. This partial transfer pathway could represent a novel mechanism of sequestering proteins within the IMS of compromised, de-energized mitochondria. It is further suggested based on experiments using an overexpressed version of Mrp17 fused to an N-terminal GFP, which blocks its import into mitochondria, that MitoTraP may normally prevent the accumulation of mistargeted MRPs in nucleoli to avoid nucleolar stress. Artificially mistargeted GFP-Mrp17 indeed interacts with several components of the 90S ribosomal pre-particle but whether mistargeting of endogenous mitochondrial ribosomal proteins can indeed ever cause nucleolar stress remains to be proven, especially since mistargeted proteins at endogenous levels might be preferentially cleared by proteasomal degradation.

While the manuscript convincingly provides evidence that a subset of mitochondrial precursor proteins is enriched in the IMS consequent to energy-poor conditions, the role of this MitoTraP pathway in shielding the cell from proteotoxicity or nucleolar stress is insufficiently substantiated, and a deeper mechanistic characterization of the retention pathway is missing. The manuscript is interesting without the potentially false claim of nucleolar mistargeting and stress. For none of the studied MRPs, evidence is presented that they, in their native form, would ever be mistargeted to nucleoli should their mitochondrial import fail. Therefore the statement that 'sequestration in the IMS prevents its mistargeting to the nucleolus and the interference with ribosome assembly', in absence of further evidence, needs to be eliminated from the abstract. The authors can speculate on such advert effects in the discussion.

To further improve upon the manuscript, the following points need to be addressed:

1. The authors claim that MitoTraP is a protective mechanism from proteotoxic effects caused by cytoplasmic accumulation of mitochondrial protein precursors, however this is not demonstrated experimentally. Further experiments should be done to determine whether interfering with MitoTraP under conditions of mitochondrial stress has deleterious effects on the cellular proteome (translation) and on the nucleolus. If these claims are retained in a final version of the manuscript, the authors need to provide the evidence for the physiological importance of the discovered mechanism.
2. It must be stated in all Figures how often the respective experiments were performed, in particular for all the experiments showing in vitro import data. These data need to be quantified from at least three independent repetitions. Lack of quantification is especially critical for the experiment presented in Fig. 4D. A band of cleaved Mrp17-cs appears to still be present in the "matrix/apyrase/proteinase K" condition, which could indicate a continued low level of import of non-conventional precursor proteins across the inner membrane of energy-deficient mitochondria.
3. It is claimed that a defined subgroup of proteins is targeted by MitoTraP, however their structural features are not completely distinguished from proteins with canonical mitochondrial targeting sequences. Both types of sequences depend on positively-charged, basic residues for import, although they are organized into amphipathic helices in canonical sequences. The authors should clarify which specific features of mitoribosomal proteins distinguish their targeting sequences from canonical ones, and whether any non-mitoribosomal proteins might also exhibit these targeting features (and should then also be subject to MitoTraP). Furthermore, the authors should discuss whether there is a subset of mitoribosomal proteins which do not contain these non-canonical sequences and test whether they are not targeted by MitoTraP. Importantly, it needs to be considered whether the preferential identification of MRPs could also be explained by a copy number effect. It is therefore recommended to also normalize the data relative to abundance.
4. In addition to the MRPs identified in Fig. 2F, many non-mitoribosomal proteins are also enriched in the IMS of Δ atp6 mutant cells. With this in mind, the authors should clarify why they chose to focus on the role of MitoTraP specifically in sequestration of MRPs rather than as a general mechanism acting on the mitochondrial proteome and whether non-MRPs scoring high for IMS trapping would be similarly affected during in vitro import. It would also be great to know how the negative control Atp1 scores as compared to other non-MRPs that were also affected (Fig. 2f). Similarly, among the MRPs, why is Mrp17 chosen for follow-up analysis? This should be elaborated on - were there unique features related to the mitochondrial targeting mechanisms of Mrp17 which made it an ideal candidate for modeling IMS retention? Why was it chosen over Mrp140 or Mrp128 (shown in Appendix Figure S4)?
5. It remains unclear which general long-term changes the Δ atp6 strain harbors as such changes might influence the state of mitochondria and import pathways. It would thus be recommended to better characterize the general state of mitochondria in this strain and to consider substantiating the data by another approach, i.e. short-term perturbation. The authors claim (Page 5) that Δ atp6 mitochondria exhibit an intact membrane potential due to the retained activity of upstream ETC complexes, however this is not demonstrated experimentally (e.g. by JC-1 staining). The ATP/ADP and NADPH/NADP⁺ ratios could also be determined as a benchmark for mitochondrial function between the wild-type and Δ atp6 mutant. Additionally, how does interfering with the ETC further upstream (with rotenone-mediated inhibition of Complex I, for example) affect mitochondrial import of e.g. Mrp17?
6. The authors state that in non-compromised mitochondria, import of Mrp17 is dependent on several Tom proteins (Fig. EV5) and Tim17, referring to Bykov et al. 2022. Does the Tim17-dependency no longer apply for MitoTraP in compromised mitochondria? This is particularly relevant in the context of the discussion, wherein the authors suggest that MRPs block import

by preventing the association of TOM-TIM23. Such a direct role for MRPs with non-conventional targeting sequences has not been shown experimentally, thus the phrasing should be adjusted.

7. While the authors demonstrate the association of GFP-Mrp17 with nucleolar pre-ribosomal particles (Fig. 5D, E), there is no evidence to claim that this association interferes with nucleolar ribosomal biogenesis or induces nucleolar stress. The authors need to show that mistargeted Mrp17 or bulk mistargeted MRPs, expressed at endogenous levels, cause a dominant-negative effect on ribosomal biogenesis.

Additional Points:

1. (Fig. EV1) The APEX constructs portrayed in Fig. EV1 do not contain GFP11, which is a discrepancy from the constructs characterized in Fig. 1B and Fig. EV3E. The authors should elaborate whether they expect there to be any key phenotypic differences between these construct variants, perhaps through the effects of GFP11-mediated retention on their localization.
2. The authors should better explain for readers outside the mitochondrial import field whether or not translocation across the outer membrane is dependent on the TIM machinery and the inner membrane gradient.
3. (Fig. 1) The authors should add a comparison between mock and IMS-APEX, akin to what is done in panels H and I, for completeness.
4. (Fig. 2) It is unclear what Erv1 is used for; is this a loading control for total mitochondria?
5. (Fig. 3) Panel H is missing. In the legend, F should be replaced with (F). Panels (E) and (F) seem redundant with each other, especially if the " $\Delta\psi$ " condition in (F) is not commented on in the text.
6. (Fig. 5C) A nucleolar marker should be included to demonstrate GFP-Mrp17 localization to the nucleolus. Based on the panels it appears that only a portion (around 50%) localizes to the nucleus while there is also a cytoplasmic pool - the authors could consider commenting on this dual localization and whether this protein could have deleterious consequences in the cytoplasm as well.
7. (Fig. EV2) The authors could also consider adding Cym1 or Atp12 as additional comparisons.
8. (Fig. EV3) C should be replaced with (C) in the legend.
9. (Fig. EV4) The wild-type samples are missing from panel C.
10. (Fig. EV6) Panel A (described in legend) is missing and panels B/C/D descriptions in legend do not align with what panels show.
11. (Fig. S1) Experiments in (B) and (C) seem redundant, authors should describe in caption why these two experiments were performed.
12. Page 7: 'Thus, Mrp17, and presumably also other MRPs, not only differ in the features of their matrix targeting sequences, but also in respect to the energy dependence of the mitochondrial import reaction (Fig. 3G)'. The conclusion on other MRPs must be deleted without further evidence.

Point-by-point response

Referee #1:

In this interesting manuscript, the authors report the novel observation that precursors of mitochondrial matrix proteins can accumulate in the intermembrane space (IMS) of ATP-depleted mitochondria. Using a proximity labeling approach, they identified proteins of the mitochondrial ribosome in the IMS of yeast mitochondria lacking the mitochondrially encoded protein ATP6. This finding was supported by a series of convincing *in vitro* experiments with isolated mitochondria. While this study provides valuable insights into the dynamics of mitochondrial protein targeting, the extent, mechanism and functional consequences of precursor accumulation in the IMS remain to be fully elucidated. At present, the authors' interpretation that the observed accumulation of mitochondrial ribosomal proteins in the IMS serves as a protective mechanism rather than a mistargeting event would require further experimental support. Alternatively, the authors could tone down this conclusion, which would not take away from the interest in their findings.

We thank the referee for the very positive evaluation. In the revised version, we considerably extended the characterization of the consequences of non-imported mitoribosomal proteins and, in addition, toned down our conclusions as suggested by the referee.

Major points:

1. To assess the extent of the IMS accumulation, the authors could quantify the fraction of mitochondrial ribosomal proteins (e.g. MRP17) that accumulate in the IMS in defective mitochondria relative to the corresponding protein level under normal conditions in the mitochondrial matrix of yeast cells. In addition, it would be informative to determine how much mitochondrial ribosomal precursor protein would accumulate outside of mitochondria if IMS accumulation is prevented (e.g., by disrupting the membrane potential)?

We followed the suggestion of the referee by two strategies. First, we purified mitochondria from $\Delta atp6$ cells and tested which fraction of MRPs became accessible to protease upon hypotonic rupturing of the outer membrane. As shown in the novel Figs. S5E and F, we found that between 10% (Mrp10) and 50% (Mrp17, Mrp136 and Mrp140) of the proteins as being protease-sensitive, confirming a considerable accumulation of several MRPs in the IMS of ATPase-deficient mutants.

Second, since the Western blot experiment was limited to proteins for which antibodies were available, we added an additional analysis document (Dataset EV3) in which the results of the APEX labeling experiment are easily accessible. Also these data suggest that considerable amounts of MRPs are present in the IMS in mitochondria of the ATPase mutants. In contrast, MRPs were barely detectable in the IMS of wild type mitochondria.

2. The authors suggest that non-canonical targeting sequences of mitochondrial proteins may mediate targeting to the IMS. To support this argument, the authors could perform comparative experiments with two (or more) mitochondrial ribosomal proteins, one containing an internal targeting sequence and the other containing a conventional N-terminal targeting sequence.

We now analyzed whether MRPs that are trapped in the IMS differ in respect to the properties of their targeting sequence from proteins that are less effected. As shown in the novel Fig. EV3F, trapped MRPs indeed tend to have a lower TargetP score, consistent with less efficient targeting information. We will follow this up in more detail in the future to identify the properties of proteins that promote their trapping in the IMS. In addition to the properties of the targeting sequence, also properties of the mature parts of the proteins (lengths, charge, structural stability, proteolytic stability) might be of relevance here. However, we felt that this analysis is beyond the scope of this study.

Moreover, in the revised version we now also added data from two additional MRPs (Mrpl40 and Mrpl28). Mrpl28 is an MRP with presequence whereas Mrpl40 lacks a cleavable presequence. We show that N-terminal GFP fusions of Mrpl40 and Mrpl28 are mistargeted to the nucleus confirming the behavior that we found for Mrp17 (novel Fig. 5E, F, EV5A, B). Moreover, import experiments showed that also Mrpl40 and Mrpl28 proteins are trapped in the IMS when ATP levels in mitochondria are low. To this end, we used the protease assay that we developed for the localization of Mrp17 with mitochondria that contained IMS- and matrix-targeted viral proteases.

3. The authors claim that non-imported mitochondrial ribosomal proteins interfere with cytosolic ribosome biogenesis. To experimentally validate this claim, the authors can quantify fully assembled ribosomes by performing a sucrose gradient fractionation experiment in yeast cells expressing a GFP tagged ribosomal protein.

The abundance of ribosomal proteins is much higher than that of MRPs. We therefore decided not to purify ribosomes on gradients, but we again used proteomics to identify the composition of ribosome-assembly complexes in the absence or presence of a competitive inhibitor of mitochondrial protein import (clogger). We generated strains, in which the 90S subunits Pwp2 and Utp18 were GFP-tagged and purified these proteins. The interactome of these assembly factors identified several MRPs when we inhibited mitochondrial protein import by expression of a clogger construct. These novel data are now shown as Figs. 6E, S7A, B, D, and Dataset EV4.

Moreover, we added data indicating the association of the non-imported GFP-Mrp17 with the 35S rRNA (novel Fig. 6D).

Finally, we used the association of ribosomal proteins with Apj1 in the nucleus. This nuclear cochaperone binds to non-assembled proteins. Upon clogger expression, many proteins of the 90S ribosome assembly complex were found as being associated with Apj1, again demonstrating that defects in mitochondrial import impair ribosome biogenesis in the nucleus (novel Fig. 5I, S7A-C, Dataset EV4).

Moreover, we downgraded the speculations about a direct interference of non-imported MRPs with the 90S complex. We agree with the referee that, at this point, it is unclear whether the effects in the nucleus arise from a direct binding of MRPs to 90S components or from more indirect effects on nuclear proteostasis.

We thank the referee for inspiring us to assess the assembly of ribosomes more thoroughly. We feel that this aspect clearly improved our study.

Minor points:

1. Figure 1B and 5B, C: Labeling of the Scale bar is too small.

We increased the scale bars as suggested.

2. Figure 3C: The presence of Proteinase K is not indicated as in the other panels of the figure.

We indicated the addition of Proteinase K.

Referee #2:

Compromised energy-generating functions of mitochondria often lead to the accumulation of non-import mitochondrial proteins, which can be toxic to the cell. In this manuscript, Flohr et al report that mitochondrial ribosomal proteins (MRPs) are sequestered in the intermembrane space (IMS) rather than accumulating outside mitochondria when the mitochondrial ATP levels, which are generally required for protein import into the matrix, are lowered. This IMS sequestration may serve a mechanism to prevent mistargeting of MRPs to the nucleus, where they could interfere with ribosome assembly. The

authors thus named this phenomenon "MitoTraP" (mitochondrial triage of precursor proteins), proposing it as a novel mechanism to reduce the toxic effects of non-imported mitochondrial proteins. The experiments were well-designed and carefully performed, and the results were analyzed properly. The findings are thought-provoking and the proposed model is worthy of consideration. Nevertheless, the idea of MitoTraP is, at least at the moment and as it stands, not convincing enough for its generality.

We appreciate the very positive evaluation of our study and addressed the critical points raised as described in the following.

The major concern is that the authors did not rule out the possibility that the observed IMS accumulation of MRPs might merely reflect dead-end products of precursor proteins at the TIM23 complex level due to compromised motor function of mitochondrial Hsp70. The authors argued that "this IMS accumulation was not simply the result of an import block of matrix proteins as the mislocalization affected only a specific subgroup of proteins", but MRP's untypical targeting signals may render them sensitive to reduced import motor function, making their translocation across the IM more vulnerable compared to precursor proteins with canonical N-terminal mitochondrial targeting sequences.

From the *in vitro* import data shown in our study, it was obvious that the import properties of Mrp17 were very different from that of 'canonical matrix proteins' such as Atp1: Mrp17 was still efficiently imported into mitochondria even when the ATP in the reaction was depleted with apyrase. Under these conditions, Atp1 was not imported at all. The same pattern was found upon inactivation of the import motor. The *in vivo* data showed that this difference is reflected by the consistent pattern of many MRPs (and some other matrix proteins) which accumulated in the IMS upon low-ATP conditions. As pointed out by the referee, this pattern is caused by an import block during protein biogenesis. However, whereas canonical matrix proteins are blocked at the level of the TOM complex so that they accumulate in the cytosol where they are degraded by the proteasome, many MRPs still traverse the TOM complex and accumulate in the IMS. Our data clearly show that the import motor is essential for protein translocation into the matrix for both groups of proteins. However, MRPs are still reaching the IMS, even if the import is blocked. We now explained this more explicitly in the revised version of our study.

The authors state that "almost the full complement of subunits of this assembly complex were co-isolated with GFP-Mrp17 (Fig. 5F)" (page 8). Then, what about other MRPs, such as Mrpl40 and Mrpl28, which also accumulate in the IMS? Further data addressing these MRPs would strengthen the study.

We now added data for the intracellular location of GFP-Mrpl40 and GFP-Mrpl28 as suggested. As shown in the novel Figs. 5E, F, EV5A, B, both proteins accumulate in the nucleus. Thus, they show the same behavior that we observed for Mrp17, supporting our conclusion that MRPs – if not imported into mitochondria – can end up in the nucleus.

The authors also demonstrate that non-imported GFP-Mrp17 is targeted into the nucleus, where it interferes with the biogenesis of 80S ribosomes. Then, does overexpression of GFP-Mrp17 indeed interfere with cytosolic ribosome assembly and negatively affect cell growth? This prediction can be experimentally tested. In relation to this, is the GFP-Mrp17 accumulation in the nucleus comparable in *atp6Δ* cells or under CCCP treatment? Clarifying these points would provide additional insights into the proposed mechanism.

We followed the suggestion of the referee. We over-expressed non-tagged Mrp17 in ATPase-deficient strains and tested the effect on cellular fitness in the ATPase-deficient mutants. Expression of this protein clearly compromised cell growth, consistent with a negative effect on ribosome assembly (novel Fig. 5H). Interestingly, over-expression of mutant version of Mrp17 that does not or less efficiently interact with the 90S complex (novel Fig. 6D) is not toxic (Fig. 5H) even though this version

also accumulates in the nucleus (novel Fig. 5G, EV5C). Thus, the expression of MRPs reduces cellular fitness under conditions at which the protein import into mitochondria is diminished, consistent with our model.

Other points.

Fig. 3. There is no explanation of DHFRmut and no description of the purpose of the experiments using the fusion proteins with DHFR and DHFRmut.

We added the explanation and the description of the purpose of this variant. DHFR^{mut} is unable to fold into a tight conformation and therefore even imported upon conditions where the motor is less active (Vestweber & Schatz, 1988. EMBO Journal 7, 1147-1151). We now extended the explanation.

Appendix Fig. S2B. Why is labeling by Matrix-APEX weaker than by IMS-APEX in *atp6Δ* cells? This may raise concerns about the reliability of APEX labeling, as low labeling efficiency could allow proteins to escape detection. The authors had better address this potential limitation and emphasize the need of caution when interpreting the results.

The signals of the streptavidin antibodies showed some variability in the Western blots owing to the sample preparation. However, we did not detect lower levels of biotinylated proteins in the matrix of $\Delta atp6$ mitochondria in the much more quantitative mass spec analysis. In order to better directly compare the APEX data of all the samples we added a novel analysis of the mass spectrometry data (novel Dataset EV3). This clearly shows that the described effects on the MRPs is not due to a lower labeling of MRPs in the matrix but a much more intense biotinylation of MRPs with the IMS-targeted APEX in the ATPase mutant mitochondria. In contrast, for other matrix proteins such as Hsp60 or Ssc1, the labeling by the matrix-APEX remained largely unaffected.

Overall, the manuscript presents intriguing findings, but further experiments and clarifications are required to substantiate the generality and robustness of the proposed MitoTraP model.

We thank for this comment and added a large set of novel data, in particular on the interference of mitochondrial protein import and ribosome assembly. See also point 3 of referee #1.

Referee #3:

Dysfunctional mitochondria often exhibit impaired capacity for protein import, leading to the extramitochondrial accumulation of mitochondrial precursor proteins, which may cause proteotoxic effects within the cell. However, the mechanisms employed by eukaryotic cells to prevent accumulation of these proteins under conditions of mitochondrial stress are not well understood. In this manuscript, the authors propose that protein retention in the intermembrane space (IMS) of damaged mitochondria, a mechanism they termed MitoTraP, is exploited by the cell to sequester mitochondrial protein precursors and prevent their extramitochondrial accumulation.

It is demonstrated through an APEX labeling approach in yeast cells that a subset of proteins containing putative non-conventional mitochondrial targeting sequences, in particular mitoribosomal proteins (MRPs), are preferentially retained in the IMS consequent to a mitochondrial defect induced by mutation of an FoF1 ATPase subunit. Using Mrp17 as a model substrate, it is demonstrated that these non-canonical proteins are transported into the IMS of compromised mitochondria independently of the presence of ATP. This partial transfer pathway could represent a novel mechanism of sequestering proteins within the IMS of compromised, de-energized mitochondria. It is further suggested based on experiments using an overexpressed version of Mrp17 fused to an N-terminal GFP, which blocks its import into mitochondria, that MitoTraP may normally prevent the accumulation of mistargeted MRPs in nucleoli to avoid nucleolar stress. Artificially mistargeted GFP-Mrp17 indeed interacts with several components of the 90S ribosomal pre-particle but whether mistargeting of endogenous mitochondrial

ribosomal proteins can indeed ever cause nucleolar stress remains to be proven, especially since mistargeted proteins at endogenous levels might be preferentially cleared by proteasomal degradation.

While the manuscript convincingly provides evidence that a subset of mitochondrial precursor proteins is enriched in the IMS consequent to energy-poor conditions, the role of this MitoTraP pathway in shielding the cell from proteotoxicity or nucleolar stress is insufficiently substantiated, and a deeper mechanistic characterization of the retention pathway is missing. The manuscript is interesting without the potentially false claim of nucleolar mistargeting and stress. For none of the studied MRPs, evidence is presented that they, in their native form, would ever be mistargeted to nucleoli should their mitochondrial import fail. Therefore the statement that 'sequestration in the IMS prevents its mistargeting to the nucleolus and the interference with ribosome assembly', in absence of further evidence, needs to be eliminated from the abstract. The authors can speculate on such advert effects in the discussion.

We thank the referee for this very positive evaluation. We tuned down the statements in the abstract as suggested and, in addition, substantiated our analysis on the impact of mistargeted mitochondrial proteins on ribosome biogenesis. We fully agree that we cannot know whether the non-imported MRPs directly bind and interfere with the 90S complex of the nucleolus or disturb the ribosome assembly machinery more indirectly. We state this now explicitly in the discussion. We also removed conclusions about the nucleolus as suggested.

To further improve upon the manuscript, the following points need to be addressed:

1. The authors claim that MitoTraP is a protective mechanism from proteotoxic effects caused by cytoplasmic accumulation of mitochondrial protein precursors, however this is not demonstrated experimentally. Further experiments should be done to determine whether interfering with MitoTraP under conditions of mitochondrial stress has deleterious effects on the cellular proteome (translation) and on the nucleolus. If these claims are retained in a final version of the manuscript, the authors need to provide the evidence for the physiological importance of the discovered mechanism.

We followed several strategies to confirm the interference of non-imported MRPs with ribosome assembly in the nucleus. First, we analyzed the interactome of Apj1 in the nucleus. This nuclear cochaperone binds to non-assembled proteins. When we prevented mitochondrial protein import by expression of a clogger construct, many proteins of the 90S ribosome assembly complex became associated with Apj1. This confirms our conclusion that defects in mitochondrial import impair ribosome biogenesis in the nucleus (novel Fig. 5I, S7C, Dataset EV4). In addition, we generated strains, in which the 90S subunits Pwp2 and Utp18 were GFP-tagged and purified these proteins. Again, the interactomes of these assembly factors were strongly influenced upon inhibition of mitochondrial protein import by clogger expression: The strong effect of clogger expression on the interactome of the 90S assembly factors supports the idea that non-imported mitochondrial precursor proteins affect ribosome biogenesis. These novel data are now shown as Figs. 6E, S7A, B, D, and Dataset EV4. Furthermore, we added data indicating the association of the non-imported GFP-Mrp17 with the 35S rRNA (novel Fig. 6D). This again supports the conclusion that MRPs can associate with 90S assembly complexes.

Nevertheless, as suggested, we downgraded the speculations about a direct interference of non-imported MRPs with the 90S complex. We agree with the referee that, at this point, it is unclear whether the effects in the nucleus arise from a direct binding of MRPs to 90S components or from more indirect effects on nuclear proteostasis.

We found that expression of Mrp17 in ATPase-deficient strains strongly reduces the growth of cells (novel Fig. 5H), however, we did not observe reduced translation rates in these strains. We show here the incorporation of ³⁵S-methionine into newly

synthesized proteins of these mutants for the inspection of the referee. We will follow these experiments up in more detail in the future.

Figure for inspection by the referee. Wild type (A, B) and $\Delta atp23$ (C, D) cells were transformed with plasmids for expression of the indicated proteins or with empty vectors (ev). Cells were grown in selective media under inducing conditions for 5 h. Cells were harvested and diluted in methionine-depleted medium. Newly synthesized proteins were radiolabeled by the addition of ^{35}S -methionine (1 $\mu\text{Ci/ml}$) for 2 or 10 min. Protein extracts were resolved by SDS-PAGE and radiolabeled proteins visualized by autoradiography.

2. It must be stated in all Figures how often the respective experiments were performed, in particular for all the experiments showing *in vitro* import data. These data need to be quantified from at least three independent repetitions. Lack of quantification is especially critical for the experiment presented in Fig. 4D. A band of cleaved Mrp17-cs appears to still be present in the "matrix/apyrase/proteinase K" condition, which could indicate a continued low level of import of non-conventional precursor proteins across the inner membrane of energy-deficient mitochondria.

We state for all quantified experiments how often they had been performed. We consistently show data from biological replicates for which samples were independently prepared (for example with independent preparations of mitochondrial fractions from the different mutants).

We also quantified the signals derived from the IMS-located proteases as suggested by the referee (corresponding to Figure 4). This nicely confirms our conclusion about the IMS-targeting of Mrp17 upon ATP depletion. This is now shown as novel Fig. 4G. Depending on the degree of ATP depletion some import into the matrix still occurred, albeit at strongly reduced levels. ATP depletion leads to a strong accumulation in the IMS of Mrp17 (as well as of Mrp128 and Mrp140) whereas these proteins were efficiently targeted into the matrix if ATP was present. This fits very well to the data from the APEX-labeling experiment which showed that in the ATPase mutants, MRPs can still be biotinylated in the matrix-APEX sample, even though often at reduced levels.

3. It is claimed that a defined subgroup of proteins is targeted by MitoTraP, however their structural features are not completely distinguished from proteins with canonical mitochondrial targeting sequences. Both types of sequences depend on positively-charged, basic residues for import, although they are organized into amphipathic helices in canonical sequences. The authors should clarify which specific features of mitoribosomal proteins distinguish their targeting sequences from canonical ones, and whether any non-mitoribosomal proteins might also exhibit these targeting features (and should then also be subject to MitoTraP). Furthermore, the authors should discuss whether there is a subset of mitoribosomal proteins which do not contain these non-canonical sequences and test whether they are not targeted by MitoTraP. Importantly, it needs to be considered whether the preferential identification of MRPs could also be explained by a copy number effect. It is therefore recommended to also normalize the data relative to abundance.

Many MRPs that are trapped in the IMS have no clear N-terminal matrix targeting sequences. As shown in the novel Fig. EV3F, trapped MRPs indeed tend to have lower TargetP scores, consistent with less efficient targeting information. We will follow this up in more detail in the future to identify the properties of proteins that promote their trapping in the IMS. In addition to the properties of the targeting sequence, also properties of the mature parts of the proteins (lengths, charge, structural stability, proteolytic stability) might be of relevance here. However, we felt that this analysis is beyond the scope of this study.

4. In addition to the MRPs identified in Fig. 2F, many non-mitoribosomal proteins are also enriched in the IMS of $\Delta atp6$ mutant cells. With this in mind, the authors should clarify why they chose to focus on the role of MitoTraP specifically in sequestration of MRPs rather than as a general mechanism acting on the mitochondrial proteome and whether non-MRPs scoring high for IMS trapping would be similarly affected during *in vitro* import. It would also be great to know how the negative control Atp1 scores as compared to other non-MRPs that were also affected (Fig. 2f). Similarly, among the MRPs, why is Mrp17 chosen for follow-up analysis? This should be elaborated on - were there unique features related to the mitochondrial targeting mechanisms of Mrp17 which made it an ideal candidate for modeling IMS retention? Why was it chosen over Mrpl40 or Mrpl28 (shown in Appendix Figure S4)?

It is true that not only MRPs are trapped in the IMS of poorly energized mitochondria. However, proteins of the large and small ribosomal subunits of mitochondria were strongly enriched in the IMS-trapped proteins (see enrichment analysis shown in Fig. 2G). Since many MRPs have unconventional targeting signals and their biogenesis was not comprehensively studied, we focused on the biogenesis of Mrp17 here, because its import properties were already studied before (Bykov et al, 2022). However, we explicitly mention in the text that also several other matrix proteins are trapped in the IMS of $\Delta atp6$ and $\Delta atp23$ cells. Atp1, the protein that we use as a non-trapped protein, was not found to accumulate in the IMS upon ATPase depletion (see Dataset EV1).

As suggested, we now show novel data for the intracellular location of GFP-Mrpl40 and GFP-Mrpl28. As shown in the novel Figs. 5E, F, EV5A, B, both proteins accumulate in the nucleus. Thus, they show the same behavior that we observed for Mrp17, supporting our conclusion that MRPs – if not imported into mitochondria – can end up in the nucleus.

5. It remains unclear which general long-term changes the $\Delta atp6$ strain harbors as such changes might influence the state of mitochondria and import pathways. It would thus be recommended to better characterize the general state of mitochondria in this strain and to consider substantiating the data by another approach, i.e. short-term perturbation. The authors claim (Page 5) that $\Delta atp6$ mitochondria exhibit an intact membrane potential due to the retained activity of upstream ETC complexes, however this is not demonstrated experimentally (e.g. by JC-1 staining). The ATP/ADP and NADPH/NADP⁺ ratios could also be determined as a benchmark for mitochondrial function between the wild-type and

$\Delta atp6$ mutant. Additionally, how does interfering with the ETC further upstream (with rotenone-mediated inhibition of Complex I, for example) affect mitochondrial import of e.g Mrp17?

As suggested by the referee, we now measured the membrane potential of the mitochondria in the $\Delta atp6$ and $\Delta atp23$ cells (novel Figs. EV2D, E, EV3D, E). These measurements confirm the conclusions that mitochondria of ATPase mutants are not as well energized as wild type mitochondria but better than mitochondria of *rho*⁰ cells that lack both the ATPase and the respiratory chain.

6. The authors state that in non-compromised mitochondria, import of Mrp17 is dependent on several Tom proteins (Fig. EV5) and Tim17, referring to Bykov et al. 2022. Does the Tim17-dependency no longer apply for MitoTraP in compromised mitochondria? This is particularly relevant in the context of the discussion, wherein the authors suggest that MRPs block import by preventing the association of TOM-TIM23. Such a direct role for MRPs with non-conventional targeting sequences has not been shown experimentally, thus the phrasing should be adjusted.

We rephrased the section of the discussion as suggested. It reads now: 'The non-conventional signals of MRPs might prevent the TOM-TIM23 association to allow trapping of the protein in the IMS, however, the mechanistic details of the translocation of MitoTraPed proteins into the IMS will have to be characterized in the future.'

7. While the authors demonstrate the association of GFP-Mrp17 with nucleolar pre-ribosomal particles (Fig. 5D, E), there is no evidence to claim that this association interferes with nucleolar ribosomal biogenesis or induces nucleolar stress. The authors need to show that mistargeted Mrp17 or bulk mistargeted MRPs, expressed at endogenous levels, cause a dominant-negative effect on ribosomal biogenesis.

As shown in the additional data that were added we show that upon clogger-mediated prevention of the MitoTraP, MRPs are found in the nucleus, associate with the co-chaperone Apj1 (novel Fig. S7C) and also induce the association of many 90S constituents with Apj1 (novel Fig. 5I). Moreover, we found several MRPs bound to Pwp2 and Utp18. See also our comments to point number 3 of referee 1.

However, we agree with the referee that we did not formally show that MRPs prevent the biogenesis of cytosolic ribosomes. We therefore tuned down the statements in the results and discussion about the physiological consequences of nucleus-targeted MRPs.

Additional Points:

1. (Fig. EV1) The APEX constructs portrayed in Fig. EV1 do not contain GFP11, which is a discrepancy from the constructs characterized in Fig. 1B and Fig. EV3E. The authors should elaborate whether they expect there to be any key phenotypic differences between these construct variants, perhaps through the effects of GFP11-mediated retention on their localization.

GFP11 is a short beta strand sequence. We used this sequence to demonstrate the accurate location of the APEX construct. In the versions used for the mass spectrometry, we did not add this GFP11 sequence. There is no reason to assume that the IMS-targeting of the IMS-APEX version depends on the presence of the GFP11 sequence.

2. The authors should better explain for readers outside the mitochondrial import field whether or not translocation across the outer membrane is dependent on the TIM machinery and the inner membrane gradient.

Protein import into the IMS occurs on two major routes: Some proteins have bipartite presequences with stop-transfer signals that are inserted by the TIM23 machinery into

the inner membrane. Upon cleavage of the targeting signals these proteins are released into the IMS. MRPs have neither transmembrane domains nor protease cleavage sites and therefore are presumably not imported by such a mechanism. Alternatively, IMS proteins are imported by assistance of the oxidoreductase Mia40 which forms intramolecular disulfide bonds. Also this does not apply to MRPs. It therefore will be interesting to study the mechanism of MRP targeting in depth in the future. We added a sentence and references to the discussion: It appears likely that their translocation mode differs from resident IMS proteins which either use bipartite presequences or embark on the MIA import pathway that depends on the oxidation of internal cysteine residues (Edwards et al. 2020 Biol Chem, Riemer et al. 2009 Science).'

3. (Fig. 1) The authors should add a comparison between mock and IMS-APEX, akin to what is done in panels H and I, for completeness.

We now added a composite PDF document in which all the signal sum values for all proteins can be easily found. This novel document is shown as Dataset EV3.

4. (Fig. 2) It is unclear what Erv1 is used for; is this a loading control for total mitochondria?

We added an explanation to the legend. Erv1 is used as loading control.

5. (Fig. 3) Panel H is missing. In the legend, F should be replaced with (F). Panels (E) and (F) seem redundant with each other, especially if the "- $\Delta\psi$ " condition in (F) is not commented on in the text.

We corrected the legend of the figure.

6. (Fig. 5C) A nucleolar marker should be included to demonstrate GFP-Mrp17 localization to the nucleolus. Based on the panels it appears that only a portion (around 50%) localizes to the nucleus while there is also a cytoplasmic pool - the authors could consider commenting on this dual localization and whether this protein could have deleterious consequences in the cytoplasm as well.

We changed the text accordingly and in the revised version, we do not claim that it localizes to nucleoli. We also show now images of Pwp2-GFP and Utp18-GFP. We agree with the referee that at this point we can not distinguish whether the GFP-Mrp17 is located in the nucleolus or in other nuclear foci. We will further characterize the location and physiological relevance of the extra-mitochondrial MRP pools in the future more comprehensively.

7. (Fig. EV2) The authors could also consider adding Cym1 or Atp12 as additional comparisons.

We now show all the APEX results for all proteins in the novel Dataset EV3, including the data for Cym1 and Atp12.

8. (Fig. EV3) C should be replaced with (C) in the legend.

We replaced C by (C).

9. (Fig. EV4) The wild-type samples are missing from panel C.

These samples are shown as Fig. 1C.

10. (Fig. EV6) Panel A (described in legend) is missing and panels B/C/D descriptions in legend do not align with what panels show.

We corrected this.

11. (Fig. S1) Experiments in (B) and (C) seem redundant, authors should describe in caption why these two experiments were performed.

We modified the figure accordingly. We had used two different wild type backgrounds, but we agree with the referee that showing one panel is sufficient.

12. Page 7: 'Thus, Mrp17, and presumably also other MRPs, not only differ in the features of their matrix targeting sequences, but also in respect to the energy dependence of the mitochondrial import reaction (Fig. 3G)'. The conclusion on other MRPs must be deleted without further evidence.

We now also added data for Mrpl28 and Mrpl40. Moreover, the APEX experiment clearly showed the accumulation in the IMS is not a unique feature of Mrp17 but a found for many MRPs.

Dear Hannes,

We have now received re-review reports from two referees, which I have included below. As you will see, you have addressed their concerns satisfactorily; however, I would like you to consider the minor changes suggested by reviewer #2. Before I can finally accept the manuscript, there are some remaining editorial points which need to be addressed. In this regard would you please:

- include the "Funding" section along with "Acknowledgements",
- include up to five keywords,
- include the "Code Availability" section along with "Data Availability",
- remove the author credit section from the manuscript,
- include callouts in the main manuscript text for figure panels 2I and 5J;
- remove callouts for missing tables: Extended Data Table 3 and 4; Extended Data Table 3,
- complete the responses in column 2 of the author checklist,
- save the legend for Dataset EV2 as a separate tab/sheet in the Excel file; include a legend for Dataset EV3,
- correct the title of the appendix to "Appendix for Dysfunctional mitochondria trap proteins in the intermembrane space",
- rename source data for 3G as source data from 3E (if this is indeed correct),
- complete the source data checklist,
- I note that a blot is reused in Figure 1D and Appendix Fig 2B, this should be acknowledged in the figure legend of Appendix Fig. 2B,
- check the nomenclature of source data files; these appear to be out by one figure,
- supply source data for Appendix Fig S5C,
- provide exact p values in the legends of figures 3B, E, G, I; 5G, 6F, G, J; 7E, F; 9E, EV3 E, F, J, K; EV4 D, F; EV5 B, C, F,
- define 'n' in the legends of figures EV5 B, C,
- use an 'Abstract' section heading,
- remove Supplemental Material from the manuscript, and
- correct the section order as follows: Title page - Abstract & Keywords - Introduction - Results - Discussion - Methods - Data Availability - Acknowledgements - Disclosure and Competing Interests Statement - References - Figure Legends - Table(s) - Expanded View Figure Legends.

I am looking forward to receiving your revised manuscript.

EMBO Press is an editorially independent publishing platform for the development of EMBO scientific publications.

Best wishes,

William

William Teale, PhD
Editor
The EMBO Journal
w.teale@embojournal.org

We realize that it is difficult to revise to a specific deadline. In the interest of protecting the conceptual advance provided by the work, we recommend a revision within 3 months (11th Aug 2025). Please discuss the revision progress ahead of this time with the editor if you require more time to complete the revisions. Use the link below to submit your revision:

Referee #1:

The revised manuscript from the group of Johannes Herrmann is substantially improved. The authors have convincingly addressed our questions with new experiments. This is an important contribution to the proteostasis field.

Referee #2:

In this revised version of the manuscript, the authors clearly addressed my concerns, presenting some substantially new results that have strengthened the manuscript. I think this manuscript reports very interesting findings that would attract the interest of a broad readership, and I recommend it for publication. The followings are a few points that the authors should consider.

The authors used "deenergized mitochondria" on page 8 to describe mitochondria lacking internal ATP, but the term "deenergized mitochondria" often means mitochondria without the membrane potential across the inner membrane. Therefore, the authors had better pay attention to using the phrase to avoid misunderstanding.

The statement "this suggests that non-imported MRPs can reduce cellular fitness" on page 9 is somewhat confusing because the non-K version of Mrp17 was also not imported into mitochondria, yet it was not toxic and did not reduce cellular fitness. The authors should avoid such a misunderstanding.

Point-by-point response

Referee #1:

The revised manuscript from the group of Johannes Herrmann is substantially improved. The authors have convincingly addressed our questions with new experiments. This is an important contribution to the proteostasis field.

We thank the referee for the very positive evaluation.

Referee #2:

In this revised version of the manuscript, the authors clearly addressed my concerns, presenting some substantially new results that have strengthened the manuscript. I think this manuscript reports very interesting findings that would attract the interest of a broad readership, and I recommend it for publication. The followings are a few points that the authors should consider.

We are very happy about this very positive and helpful evaluation of our study and addressed the two small remaining points as described in the following.

1. The authors used "deenergized mitochondria" on page 8 to describe mitochondria lacking internal ATP, but the term "deenergized mitochondria" often means mitochondria without the membrane potential across the inner membrane. Therefore, the authors had better pay attention to using the phrase to avoid misunderstanding.

We rewrote the sentence which reads now: 'This indeed suggested that Mrp17 was transported into the IMS when mitochondrial ATP levels were low.'

2. The statement "this suggests that non-imported MRPs can reduce cellular fitness" on page 9 is somewhat confusing because the non-K version of Mrp17 was also not imported into mitochondria, yet it was not toxic and did not reduce cellular fitness. The authors should avoid such a misunderstanding.

We deleted this sentence as it was anyway largely redundant with the sentence before.

Dear Hannes,

I am pleased to inform you that your manuscript has been accepted for publication in the EMBO Journal.

Congratulations!

Best wishes,

William

William Teale, PhD
Editor
The EMBO Journal
w.teale@embojournal.org
